# PMCA-generated prions from the olfactory mucosa of patients with Fatal Familial Insomnia cause prion disease in mice

Edoardo Bistaffa[1†], Alba Marín-Moreno[2†], Juan Carlos Espinosa[2], Chiara Maria Giulia De Luca[1,3], Federico Angelo Cazzaniga[1], Sara Maria Portaleone[4], Luigi Celauro[3], Giuseppe Legname[3], Giorgio Giaccone[1], Juan Maria Torres[2], Fabio Moda[1]*

[1]Fondazione IRCCS Istituto Neurologico Carlo Besta, Division of Neurology 5 and Neuropathology, Milan, Italy; [2]Centro de Investigación en Sanidad Animal (CISA-INIA), Valdeolmos, Madrid, Spain; [3]Scuola Internazionale Superiore di Studi Avanzati (SISSA), Department of Neuroscience, Laboratory of Prion Biology, Trieste, Italy; [4]ASST Santi Paolo e Carlo, Department of Health Sciences, Otolaryngology Unit, Università Degli Studi di Milano, Milan, Italy

## Abstract

**Background:** Fatal Familial Insomnia (FFI) is a genetic prion disease caused by the D178N mutation in the prion protein gene (PRNP) in coupling phase with methionine at PRNP 129. In 2017, we have shown that the olfactory mucosa (OM) collected from FFI patients contained traces of PrPSc detectable by Protein Misfolding Cyclic Amplification (PMCA).

**Methods:** In this work, we have challenged PMCA-generated products obtained from OM and brain homogenate of FFI patients in BvPrP-Tg407 transgenic mice expressing the bank vole prion protein to test their ability to induce prion pathology.

**Results:** All inoculated mice developed mild spongiform changes, astroglial activation, and PrPSc deposition mainly affecting the thalamus. However, their neuropathological alterations were different from those found in the brain of BvPrP-Tg407 mice injected with raw FFI brain homogenate.

**Conclusions:** Although with some experimental constraints, we show that PrPSc present in OM of FFI patients is potentially infectious.

**Funding:** This work was supported in part by the Italian Ministry of Health (GR-2013-02355724 and Ricerca Corrente), MJFF, ALZ, Alzheimer's Research UK and the Weston Brain Institute (BAND2015), and Euronanomed III (SPEEDY) to FM; by the Spanish Ministerio de Economía y Competitividad (grant AGL2016-78054-R [AEI/FEDER, UE]) to JMT and JCE; AM-M was supported by a fellowship from the INIA (FPI-SGIT-2015-02).

*For correspondence:
Fabio.Moda@istituto-besta.it

†These authors contributed equally to this work

Competing interests: The authors declare that no competing interests exist.

## Introduction

Fatal Familial Insomnia (FFI) is a genetic prion disorder caused by the conformational conversion of the cellular form of the prion protein (PrP$^C$) into an abnormally folded conformer, named prion or PrP$^{Sc}$, which acquires toxic properties and accumulates in the brain (*Lugaresi et al., 1986*). In 1992, it was shown that the cause of the disease is an autosomal-dominant mutation at codon 178 of the PrP$^C$ encoding gene (*PRNP*), resulting in aspartic acid (D) to asparagine (N) substitution (D178N) (*Medori et al., 1992*). Since a main clinical feature of the disease was the presence of a severe and

untreatable insomnia, it was named FFI. The pathology primarily affects the thalamus that shows neuronal loss, spongiform changes, astrogliosis, and moderate PrP$^{Sc}$ deposition (*Montagna et al., 2003*; *Macchi et al., 1997*). In cases of long disease duration, such alterations involve also the cerebral cortex. The polymorphism at codon 129 of the *PRNP* (methionine [M] or valine [V]) plays a significant role in disease heterogeneity (*Monari et al., 1994*). For instance, the presence of MM (FFI$^{D178N-129MM}$) causes FFI with rapid progression (7–18 months), sleep disturbances, and dysautonomia. The presence of MV (FFI$^{D178N-129MV}$) is associated with longer disease duration (20–35 months) with equilibrium and gait dysfunctions, while the presence of VV causes a distinct genetic disease, referred to as familial Creutzfeldt–Jakob disease (fCJD$^{D178N-129VV}$) (*Gambetti et al., 1995*). Several lines of evidence suggest that codon 129 polymorphism in *PRNP* does not influence the age at disease onset (*Montagna et al., 2006*). PrP$^C$ is synthesized in the rough endoplasmic reticulum where it undergoes several post-translation modifications, including the addition of a GPI anchor to its C-terminal part, the formation of a disulfide bridge between two cysteine residues (Cys179-Cys214), and the N-linked glycosylation at two asparagine residues (Asn181 and Asn197) (*Turk et al., 1988*). Finally, in the Golgi apparatus, the oligosaccharides are modified to produce complex-type chains rich in sialic acid (*Caughey et al., 1989*), which have an important role in targeting PrP$^C$ to neuronal synapses (*Bate et al., 2016*). The pattern of glycosylation gives rise to different PrP$^C$ forms: the di-glycosylated (70%), the mono-glycosylated (25%), and the un-glycosylated (5%) species (*Caughey et al., 1989*; *Rudd et al., 1999*). All PrP$^C$ species are soluble in detergent and sensitive to proteinase K (PK) digestion. The D178N mutation reduces the structural stability of the PrP$^C$, which is more prone to convert into PrP$^{Sc}$ and aggregate (*Liemann and Glockshuber, 1999*). This latter is less soluble in detergent and is partially resistant to PK digestion. After PK treatment, the resistant un-glycosylated band of PrP$^{Sc}$ associated with FFI$^{D178N-129MM}$ or FFI$^{D178N-129MV}$ shows a molecular weight of 19 kDa, while that associated with fCJD$^{D178N-129VV}$ has a molecular weight of 21 kDa (*Monari et al., 1994*). These differences suggest that the mutation at position 178 and the polymorphism at 129 of the *PRNP* affect the tertiary structure of the protein, thus influencing the cleavage of PK at the N-terminal region. Such alterations in PrP$^{Sc}$ structure might be the factor responsible for the phenotypic heterogeneity of FFI. The ability of the D178N mutation to promote conversion of PrP$^C$ into PrP$^{Sc}$ that results in the generation of spontaneous prion pathology was shown in mice genetically modified to express murine PrP with the D177N mutation (the equivalent to human D178N) where PrP$^{Sc}$ is barely detectable (*Bouybayoune et al., 2015*; *Jackson et al., 2009*). The infectious properties of FFI-PrP$^{Sc}$ were extensively investigated by transmission studies in wild-type animals (*Tateishi et al., 1995*; *Sasaki et al., 2005*) and in several lines of mice genetically modified to express human PrP (*Telling et al., 1994*; *Telling et al., 1995*) as well as primates (*Tateishi et al., 1995*; *Brown et al., 1994*; *Collinge et al., 1995*; *Telling et al., 1996*; *Takeuchi et al., 2019*). Results of these studies showed that several factors, including the polymorphisms at codon 129 of the incoming FFI-PrP$^{Sc}$ inoculum, the prion titer of the FFI brain areas used to challenge the animals, and the PrP$^C$ amino acid sequence of the recipient animals, could significantly influence the transmission efficiency. Notably, when inoculated in mice expressing a chimeric human-mouse PrP, FFI-PrP$^{Sc}$ is able to promote the generation of prions characterized by similar biochemical features of the original inoculum (*Telling et al., 1996*). In a recent report, FFI was efficiently transmitted to knock-in mice expressing the bank vole PrP (BvPrP) with methionine at codon 109 of the PrP$^C$ (*Takeuchi et al., 2019*). Also in this case, the PrP$^{Sc}$ generated in mice showed the typical biochemical features of FFI-PrP$^{Sc}$ (predominance of the di-glycosylated band and migration of the un-glycosylated PrP species at 19 kDa). By exploiting the recently developed Protein Misfolding Cyclic Amplification (PMCA) (*Saborio et al., 2001*) and Real-Time Quaking-Induced Conversion (RT-QuIC) assays (*Atarashi et al., 2011*), it was shown that FFI-PrP$^{Sc}$ can promote PrP$^C$ misfolding and aggregation *in vitro* (*Orrú et al., 2015*). In this context, we have previously shown that the PMCA is able to detect traces of PrP$^{Sc}$ present in the olfactory mucosa (OM) of a patient with FFI (at a late stage of the disease) using the brain of bank voles (Bv109M) as a reaction substrate (*Redaelli et al., 2017*).

In the present study, we verified whether PMCA-generated products possess infectious properties when challenged in mice genetically modified to express the BvPrP with methionine at codon 109 (BvPrP-Tg407) (*Espinosa et al., 2016*). In particular, animals were inoculated with (1) raw FFI or Alzheimer's disease (AD) brain homogenates (FFI-BH or AD-BH), (2) their PMCA products (FFI-BH_PMCA or AD-BH_PMCA), and (3) PMCA products obtained from OM of FFI or AD patients (FFI-

OM_PMCA or AD-OM_PMCA). Our results indicate that BvPrP-Tg407 mice are highly susceptible to infection by FFI-BH but also by the PMCA products (FFI-BH_PMCA and FFI-OM_PMCA). Particularly, animals inoculated with PMCA products developed prion isolates and neuropathological changes significantly different from those observed in mice challenged with FFI-BH. It is likely that, through PMCA, we have generated bank vole-adapted FFI prion isolates that acquired infectious properties distinct from that of the original FFI, and this resulted in the appearance of different pathological phenotypes of the disease. As expected, none of the animals inoculated with AD-related samples developed prion pathology.

# Materials and methods

## Key resources table

| Reagent type (species) or resource | Designation | Source or reference | Identifiers | Additional information |
|---|---|---|---|---|
| Strain, strain background (mouse) | BvPrP-Tg407 mice | DOI:10.1128/JVI.01592-16 | | |
| Peptide, recombinant protein | recBvPrP$_{90-231}$ | DOI:10.1038/srep46269 | | |
| Antibody | 6D11 | Covance | Catalog # SIG-399810 RRID:AB_2564735 | WB: 0.2 µg/mL |
| Antibody | Sha31 | SPI Bio | Catalog # A03213 | WB: 0.4 µg/mL |
| Antibody | Mouse IgG HRP Linked F(ab')2 Fragment | GE | Catalog # NA9310V | |
| Antibody | Saf34 | | | Gently provided by Prof. Jacques Grassi DOI:10.1111/j.1471-4159.2004.02356.x DOI:10.1006/bbrc.1999.1730 IHC: 1.25 µg/mL |
| Antibody | GFAP antibody | DAKO | Catalog # Z0334 RRID:AB_10013382 | IHC: 0.5 µg/mL |
| Chemical compound, drug | PBS | Gibco | Catalog # 14200-067 | |
| Chemical compound, drug | Isofluorane | Isoba vet Schering-Plough S.A. (Merck company) | Catalog # 792632 | |
| Chemical compound, drug | Glucose | Gibco | Catalog # A2494001 | |
| Chemical compound, drug | Proteinase K | Invitrogen | Catalog # AM2542 | |
| Chemical compound, drug | Guanidine hydrochloride | Sigma-Aldrich | Catalog # 50950 | |
| Chemical compound, drug | NaCl | Carlo Erba | Catalog # 479686 | |
| Chemical compound, drug | EDTA | Sigma-Aldrich | Catalog # 03690 | |
| Chemical compound, drug | NP40 | BDH | Catalog # 56009 | |
| Chemical compound, drug | Deoxycholic acid, sodium salt | Millipore | Catalog # 264101 | |
| Chemical compound, drug | Tris-hydroxymethyl-aminomethane (Tris-HCl) | Carlo Erba | Catalog # 489973 | |
| Chemical compound, drug | Blotto, non-fat dry milk | Santa Cruz | Catalog # SC-2325 | |

*Continued on next page*

| Reagent type (species) or resource | Designation | Source or reference | Identifiers | Additional information |
|---|---|---|---|---|
| Chemical compound, drug | N-Lauroylsarcosine sodium salt (sarkosyl) | Sigma-Aldrich | Catalog # 61747 | |
| Chemical compound, drug | Phosphate buffered saline -RT-QuIC | Sigma-Aldrich | Catalog # P5493 | |
| Chemical compound, drug | Sodium dodecyl sulfate solution | Sigma-Aldrich | Catalog # 71736 | |
| Chemical compound, drug | Thioflavin T | Sigma-Aldrich | Catalog # T3516 | |
| Chemical compound, drug | Ethanol (for Carnoy fixative preparation) | Sigma-Aldrich | Catalog # 32221 | DOI:10.1111/j.1750-3639.2000.tb00240.x |
| Chemical compound, drug | Acetic acid glacial (for Carnoy fixative preparation) | Sigma-Aldrich | Catalog # 33209 | DOI:10.1111/j.1750-3639.2000.tb00240.x |
| Chemical compound, drug | Paraffin | Bio Optica | Catalog # 08-7920 | |
| Chemical compound, drug | Chloroform (for Carnoy fixative preparation) | Carlo Erba | Catalog # 438614 | DOI:10.1111/j.1750-3639.2000.tb00240.x |
| Chemical compound, drug | Guanidine thiocyanate | Merck-Millipore | Catalog # 1.04167.0250 | |
| Commercial assay or kit | Bolt 12%, Bis-Tris, 1.0 mm, Mini Protein Gel, 10-well | Invitrogen | Catalog # NW00120BOX | |
| Commercial assay or kit | Immobilon-P, PVDF, 0.45 µm | Millipore | Catalog # IPVH00010 | |
| Commercial assay or kit | 4X Bolt LDS Sample Buffer | Invitrogen | Catalog # B0007 | |
| Commercial assay or kit | 10X Bolt Sample Reducing Agent | Invitrogen | Catalog # B0009 | |
| Commercial assay or kit | ECL Prime Western Blotting System | Amersham | Catalog # RPN2232 | |
| Commercial assay or kit | ARK (Animal Research Kit) | DAKO | Catalog # K3954 | |
| Commercial assay or kit | Liquid DAB Substrate Chromogen System | DAKO | Catalog # K3468 | |
| Commercial assay or kit | PNGase F | New England Biolabs | Catalog # P0704S | |
| Other | Nanosep Centrifugal Devices | Pall Corporation | Catalog # OD100C34 | |
| Other | Hematoxylin | Bio Optica | Catalog # 05-06012 | |
| Other | Eosin | Bio Optica | Catalog # 05-10002 | |
| Other | Thioflavin S | Sigma-Aldrich | Catalog # T-1892 | |
| Software, algorithm | ImageJ | PMID:22743772 | RRID:SCR_003070 | |
| Software, algorithm | GraphPad PRISM 5.0 v | N/A | RRID:SCR_002798 | |
| Software, algorithm | Nikon ACT-1 acquisition software | N/A | https://www.nikon.com/products/microscope-solutions/support/download/software/camerasfor/act1_v263.htm | |
| Software, algorithm | G:BOX Chemi XT4 | N/A | http://www.alphametrix.de/page/index.php?category=geldoc&pageid=219 | |

## BvPrP-Tg407 animal model

BvPrP-Tg407 transgenic mice were used for bioassay of the FFI and AD products as previously described. These mice express physiological levels of bank vole PrP$^C$ with methionine at codon 109 in the absence of murine PrP$^C$ (*Espinosa et al., 2016*).

## Preparation and analysis of the inocula

Frozen samples of frontal cortex collected from a patient with FFI$^{D178N-129MM}$ or a patient with AD homozygous for methionine at 129 *PRNP* codon were homogenized at 10% (weight/volume; w/v) in PBS. These samples were named FFI-BH and AD-BH. Both specimens were subjected to three rounds of PMCA (as described in *Redaelli et al., 2017*) and named FFI-BH_PMCA or AD-BH_PMCA, respectively. OM samples collected from a patient with FFI$^{D178N-129MM}$ and a patient with AD (*PRNP*-129MM) were named FFI-OM and AD-OM. These samples were subjected to three rounds of PMCA (see *Redaelli et al., 2017*) and named FFI-OM_PMCA and AD-OM_PMCA, respectively. All PMCA reactions were performed using the brain of bank vole homozygous for methionine at codon 109 of the *Prnp* (Bv109M).

All samples were treated with PK and subjected to Western blot (Wb) analysis to check for the presence of PrP$^{Sc}$. To this aim, FFI-BH and AD-BH and all PMCA-generated products (FFI-BH_PMCA, FFI-OM_PMCA, AD-BH_PMCA, and AD-OM_PMCA) were thawed and analyzed without the need of pretreatments, while FFI-OM and AD-OM were thawed and sonicated for 1 min before the analysis.

For the inoculation, (1) FFI-BH and AD-BH did not need any pretreatment; (2) FFI-OM and AD-OM were not challenged because of the lack of material (see BvPrP-Tg407 mouse bioassay section); and (3) FFI-BH_PMCA, FFI-OM_PMCA, AD-BH_PMCA, and AD-OM_PMCA were centrifuged at high speed (100,000 × g) for 1 hr at 4°C and final pellets were suspended in the same volume of PBS before the injections.

## BvPrP-Tg407 mouse bioassay

Animals have been divided into six groups according to the inoculum: (1) FFI-BH (n = 6), (2) FFI-BH_PMCA (n = 5), (3) FFI-OM_PMCA (n = 7), (4) AD-BH (n = 4), (5) AD-BH_PMCA (n = 6), and (6) AD-OM_PMCA (n = 5). For the inoculations, animals from 6- to 7-week-old were intracerebrally inoculated after isoflurane anesthesia (Isoba vet Schering-Plough S.A.) with 20 µL of the different inocula that were placed in the right parietal lobe by using a 25-gauge disposable hypodermic needle. After inoculation, mice were checked twice a week. When progression of prion disease was evident or at the end of mice lifespan (650 dpi), mice were sacrificed and the survival time was calculated. Brains were collected and divided into two parts: one dedicated to biochemical analysis and the other dedicated to histological analysis (when possible).

Unfortunately, we could not inoculate FFI-OM because the sample was not enough and we did not have time to recollect it before the death of the patient.

## Preparation of BvPrP-Tg407 brains for biochemical analysis

Half of the brain collected from BvPrP-Tg407-inoculated mice was homogenized at 10% (w/v) in 5% glucose and subjected to Wb for testing the presence of PK-resistant PrP (PrP$^{res}$). In particular, the samples were digested with 50 µg/mL of PK (1 hr, 37°C, 550 rpm; Invitrogen) before Wb analysis. To verify the migration pattern of the un-glycosylated PrP band, samples were treated with PNGase F. To better characterize the biochemical properties of the PrP$^{res}$, samples were subjected to PK-resistance assay, conformational stability assay, and RT-QuIC analysis. As PrP$^{res}$ migration controls, 10% (w/v) brain homogenates from (1) FFI, (2) sporadic Creutzfeldt–Jakob disease homozygous for methionine at codon 129 and type 1 PrP$^{res}$ (sCJD-129MM1), and (3) AD patients were prepared in lysis buffer (100 mM NaCl, 10 mM EDTA, 0.5% NP-40, 0.5% sodium deoxycholate, 10 mM Tris-HCl, pH 7.4), digested with 50 µg/mL of PK (1 hr, 37°C, 550 rpm; Invitrogen) and analyzed by Wb.

## Wb analysis

Samples were subjected to Wb analysis using 12% Bis-Tris plus gels (ThermoScientific) and then transferred into polyvinylidene difluoride membranes (PVDF, Millipore). Before the electrophoretic separation, samples were boiled at 100°C for 10 min in loading buffer (Bolt LDS Sample Buffer and

DTT, ThermoScientific). After blocking with non-fat dry milk (1 hr at room temperature), the membranes were probed with the monoclonal anti-PrP antibodies 6D11 (which recognizes the N-terminal part of the PrP [a.a. 93–109]; 0.2 µg/mL – Covance SIG-399810) or Sha31 antibody (which recognizes a more C-terminal region of the PrP [a.a. 145–152]; 0.4 µg/mL – SPI Bio a03213). After incubation with secondary antibody (Fab fragment anti-mouse IgG conjugated with horseradish peroxidase [HRP], GE), membranes were developed using the ECL Prime detection system (Amersham) and chemiluminescence was visualized using a G:BOX Chemi Syngene system.

## PK-resistance assay

10 µL of (1) FFI-BH (10-fold concentrated by means of high-speed centrifugation), (2) FFI-BH_PMCA, and (3) brain homogenates of BvPrP-Tg407-inoculated mice were treated with five increasing concentrations of PK (50, 100, 250, 500, and 1000 µg/mL) and incubated for 1 hr at 37°C under shaking (500 rpm). The enzymatic activity was stopped by boiling the samples at 100°C for 10 min in loading buffer (Bolt LDS Sample Buffer and DTT, ThermoScientific). Samples were subjected to Wb analyses and membranes immunoblotted with the 6D11 antibody. Resulting PK-resistant PrP bands were subjected to three independent measurements, and densitometric quantification was performed using the ImageJ software.

## Conformational stability assay

50 µL of (1) FFI-BH (10-fold concentrated by means of high-speed centrifugation), (2) FFI-BH_PMCA, and (3) brain homogenates of BvPrP-Tg407-inoculated mice were treated with 450 µL of guanidine hydrochloride (Gdn-HCl; Sigma) at different molar concentrations (1.5, 3, 4.5, and 6 M) for 2 hr at 25°C under shaking (550 rpm). Subsequently, an equal volume of sarkosyl 20% (Sigma) was added to the preparation that was incubated for 10 min with gentle shaking. Samples were centrifuged at $100,000 \times g$ for 1 hr at 4°C. Pellets were washed with PBS and then centrifuged at high speed ($100,000 \times g$, 30 min at 4°C). The resulting pellets were suspended in 50 µL of loading buffer (Bolt LDS Sample Buffer and DTT, ThermoScientific) and then subjected to Wb analysis as described. Membranes were immunoblotted using the 6D11 antibody. Each densitometric analysis of the resulting bands was performed (at least three times per sample) using ImageJ software.

## PNGase F analysis

PNGase experiments were performed according to the manufacturer's protocol (New England Biolabs PNGase F P0704S). Briefly, 10 µL of BvPrP-Tg407 brain homogenates was treated with PK (50 µg/mL, 1 hr at 37°C) and the digestion was stopped by the addition of 2 µL of Denaturing Buffer (10X) followed by boiling at 100°C for 10 min. Subsequently, the samples were supplemented with 2 µL of PNGase F, 2 µL of NP-40, 2 µL of GT buffer, and 2 µL of PBS $1\times$. The samples were incubated overnight at 37°C under continuous shaking (550 rpm). Samples were then analyzed by Wb and membranes immunoblotted with the 6D11 antibody. As controls for PrP$^{res}$ migration, brain homogenates of FFI (un-glycosylated PrP migrating at 19 kDa), sCJD-129MM1 (un-glycosylated PrP migrating at 21 kDa), and AD (negative control) were used.

## RT-QuIC analysis

Before RT-QuIC analysis, brain homogenates of all BvPrP-Tg407-inoculated mice were diluted at $10^{-3}$ (volume/volume) in PBS (Gibco). As a reaction substrate, the recombinant truncated BvPrP with methionine at codon 109 (recBvPrP$_{90-231}$) was prepared and used as already described (*Redaelli et al., 2017*). The substrate was allowed to thaw at room temperature and filtered through a 100 kDa Nanosep centrifugal device (Pall Corporation). The final reaction mix was composed by 10 mM PBS, 1 mM EDTA, 150 mM NaCl, 0.002% SDS, 10 µM Thioflavin T, and 0.13 mg/mL of recBvPrP$_{90-231}$. 2 µL of each diluted brain sample was added to 98 µL of reaction mix. Every sample was analyzed at least three times in triplicate in a 96-well optical flat bottom plate (ThermoScientific) that was inserted into a FLUOstar OPTIMA microplate reader (BMG Labtech) and subjected to cycles of shaking (1 min, 600 rpm, double orbital) and incubation (1 min) at 55°C. Fluorescence readings (480 nm) were taken every 15 min (450 nm, 30 flashes per well). A sample was considered positive if the mean of the highest two fluorescence values (AU) of the replicates was higher than 10,000 AU and at least two out of three replicates crossed this threshold before 18 hr. Data were plotted in a

graph showing the time taken for each replicate (black dots) to reach the fluorescence threshold (lag phase).

### Neuropathological analysis

Half of the brain collected from BvPrP-Tg407 mice was fixed in Carnoy solution and embedded in paraffin (*Giaccone et al., 2006*). 7-µm-thick serial sections were stained with hematoxylin and eosin (H&E) and thioflavin S, or immunostained with monoclonal antibodies to PrP (Saf34; 1.25 µg/mL; a.a. 59–89, gently provided by Prof. Jacques Grassi), and polyclonal antibodies to glial fibrillary acidic protein (GFAP; 0.5 µg/mL; Dako Z0334). Before PrP immunostaining, sections were treated with PK (5 µg/mL, 5 min, room temperature) and guanidine isothiocyanate (3 M, 20 min, room temperature). Non-specific bindings of the primary antibody were prevented using the ARK kit (Dako). Reactions were visualized using the 3–3′ diaminobenzidine (DAB, Dako) as chromogen. Samples were analyzed under a Nikon Eclipse E800 microscope equipped with a Nikon digital camera DXM 1200 and Nikon ACT-1 (v2.63) acquisition software.

### Statistical analysis

Log-rank test was used for the analysis of the survival time. Two-way ANOVA followed by Bonferroni post-tests was used for p calculation (* ° ♦ □$p < 0.05$, ** °° ♦♦ □□$p < 0.01$, *** °°° ♦♦♦ □□□$p < 0.001$). Mean values are presented with their standard errors of the mean (SEM). Statistical analysis and graphic representations were performed with Prism software (v. 5.0 GraphPad). Densitometric analysis was performed using ImageJ software (v. 1.48). Samples analyzed by RT-QuIC were considered positive if the mean of the highest two fluorescence values (AU) of the replicates was higher than 10,000 AU and at least two out of three replicates crossed this threshold before 18 hr.

## Results

### Biochemical analysis of the inocula

Before injections, all inocula were subjected to Wb analysis to evaluate the presence of PrP$^{res}$ (*Figure 1*). Notably, the prions observed in FFI-BH, FFI-BH_PMCA, and FFI-OM_PMCA showed analogous biochemical properties that were characterized by a predominance of the di-glycosylated PrP band and the migration of un-glycosylated one at 19 kDa. As a control of migration, we have included the brain homogenate of a patient with sCJD-129MM1 (sCJD-BH T1) that is instead characterized by a PrP$^{res}$ with high levels of the mono-glycosylated PrP species with the un-glycosylated one migrating at 21 kDa. As expected, we did not detect PrP$^{res}$ in FFI-OM sample before amplification. None of the samples collected from patients with Alzheimer's disease showed a PrP$^{res}$ signal. Asterisks indicate samples that were not inoculated because of the lack of material (see BvPrP-Tg407 mouse bioassay section in Materials and methods).

### Survival time

All animals inoculated with FFI-BH, FFI-BH_PMCA, and FFI-OM_PMCA (continuous lines in *Figure 2*) succumbed to prion disease, while those inoculated with AD-BH, AD-BH_PMCA, and AD-OM_PMCA did not and were sacrificed at the end of the experiment (dashed lines in *Figure 2*). The clinical presentation of the disease was similar between groups and mainly characterized by the presence of typical prion signs and symptoms including, ataxia, generalized tremor, sustained hunched posture, and extensive piloerection. Although animals inoculated with FFI-BH showed clinical alterations earlier (431 ± 58 dpi) than those inoculated with FFI-BH_PMCA (489 ± 13 dpi) and FFI-OM_PMCA (474 ± 27 dpi), the differences in the survival times did not reach a statistical significance (p=0.085, log-rank test) (*Figure 2*).

### Biochemical analysis of prions generated in BvPrP-Tg407-inoculated mice

Five out of six animals of the FFI-BH group developed a PrP$^{res}$ mainly characterized by a predominance of the di-glycosylated band with the un-glycosylated one migrating at 19 kDa. Unexpectedly, in one of these animals (the number #5), we have detected a PrP$^{res}$ characterized by a predominance of the mono-glycosylated band with the un-glycosylated one migrating at 19 kDa. These findings

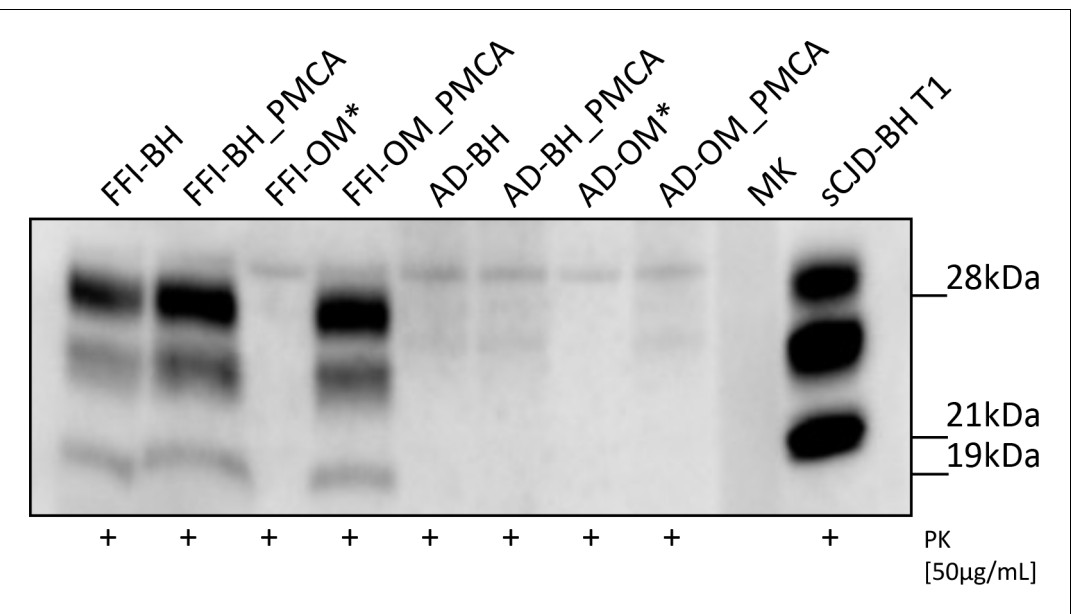

**Figure 1.** Western blot analysis of the inocula. Prions were detected in all Fatal Familial Insomnia (FFI) samples except for the raw olfactory mucosa (FFI-OM). Notably, the glycoform ratio of all PrPres was identical and characterized by a predominance of the di-glycosylated band with the un-glycosylated one migrating at 19 kDa. No PrPres was found in samples collected from patients with AD. Three rounds of Protein Misfolding Cyclic Amplification (PMCA) were performed for each sample (brain homogenate [BH] or OM) before the analysis. BH of a sCJD-129MM1 patient (sCJD-BH T1) was used as migration control. Asterisks indicate samples that were not inoculated in mice. MK: molecular weight marker. Samples were immunoblotted with anti-PrP 6D11 antibody.

were confirmed using either the 6D11 antibody or the Sha31 antibody (*Figure 3a*, *Figure 3—figure supplement 1*). This suggests that a different prion might have emerged in this mouse. Therefore, we have decided to remove this animal from the group and analyze it separately. Moreover, the last mouse of this group (the number #6) that was sacrificed at 605 dpi showed lower PrPres signal (*Figure 3a, Figure 3—figure supplement 1*) compared to all the other animals sacrificed at earlier time points. All the animals inoculated with FFI-BH_PMCA showed a PrPres signal characterized by a predominance of the di-glycosylated band with the un-glycosylated one migrating at 19 kDa (*Figure 3b*). Similar findings were observed in the brains of animals inoculated with FFI-OM_PMCA (*Figure 3c*). These results were confirmed using both the 6D11 and the Sha31 antibodies (*Figure 3—figure supplement 1*). However, animal number #1 of the FFI-OM_PMCA group was characterized by the presence of two faint un-glycosylated PrPres fragments detectable only with the use of the N-terminal 6D11 antibody. This difference could not be appreciated with the use of the C-terminal Sha31 antibody where a single un-glycosylated band migrating at 19 kDa was detected (*Figure 3—figure supplement 1*). Thus, almost all PMCA-inoculated mice developed a PrPres with the typical glycoform profile of the FFI prion. However, it should be kept in mind that, at different levels, the human FFI prions present in BH and OM have been subjected to several analyses performed using the bank vole PrPC (either as PMCA reaction substrate or as transgenic mice). In particular, the original FFI strains have been subjected to PMCA analyses with bank vole brain (Bv109M) (see *Redaelli et al., 2017*) and these reaction products as well as the raw FFI-BH were inoculated in animals that express the BvPrP (BvPrP-Tg407). Therefore, even if some biochemical features of the human FFI have been retained by the prions generated in these mice, they represent a bank vole-adapted version of the original FFI strain. To better analyze the migration profile of the un-glycosylated PrP bands, we have performed PNGase F experiments and observed that all prions generated in the brain of BvPrP-Tg407 mice migrated at 19 kDa, regardless of the inoculum (*Figure 3d*). Also in the case of animal number #1 of the FFI-OM_PMCA group, the PNGase F treatment of the sample resulted in the appearance of a single un-glycosylated band migrating at 19 kDa. No PrPres was found in the brain of BvPrP-Tg407 mice inoculated with AD-BH, AD-BH_PMCA, and AD-OM_PMCA.

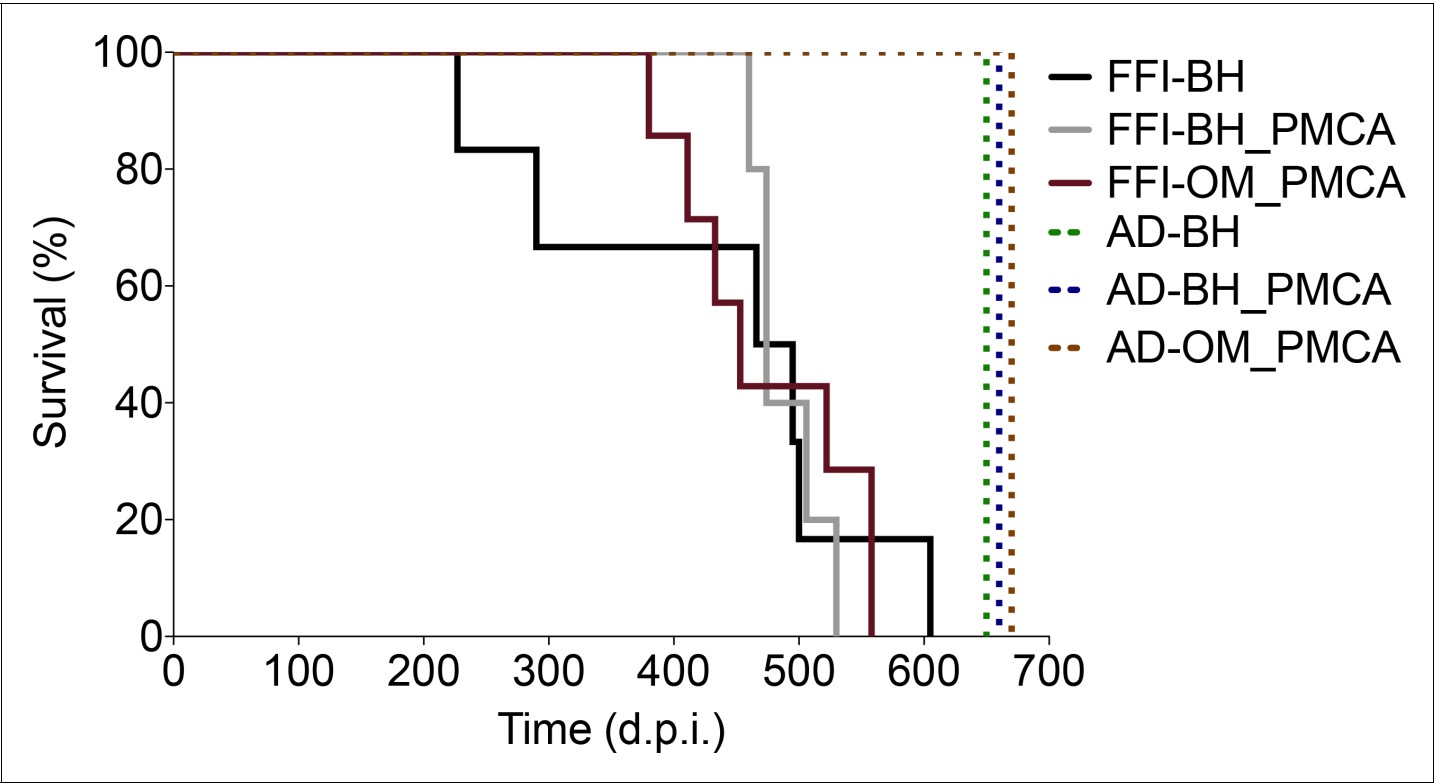

**Figure 2.** Survival time. All animals inoculated with FFI-BH, FFI-BH_PMCA, and FFI-OM_PMCA succumbed to prion disease with similar survival time (431 ± 58 dpi, 489 ± 13 dpi, and 474 ± 27 dpi, respectively; p=0.085, log-rank test), while those inoculated with AD-BH, AD-BH_PMCA, and AD-OM_PMCA did not and were sacrificed at the end of the experiment. FFI: Fatal Familial Insomnia; PMCA: Protein Misfolding Cyclic Amplification; OM: olfactory mucosa; BH: brain homogenate.

### Evaluation of PK-resistance and conformational stability of prions generated in BvPrP-Tg407-inoculated mice

Brain homogenates of BvPrP-Tg407 were treated with increasing concentrations of PK. As shown in *Figure 4*, prions found in the brains of animals inoculated with FFI-BH were significantly more resistant to proteolytic digestion than those found in the brains of animals inoculated with either FFI-BH_PMCA or FFI-OM_PMCA (*Figure 4a*). As already mentioned, we decided to separate animal number #5 from the FFI-BH group because of the divergent glycoform profile of its PrP$^{res}$. Moreover, this prion was less resistant to PK digestion and more stable towards Gdn-HCl treatment compared to all the others of the same group (*Figure 4—figure supplement 1*). Conversely, the PK-resistance profile of the PrP$^{res}$ found in animals inoculated with FFI-BH_PMCA and FFI-OM_PMCA was comparable. This suggests that the PMCA might have favored the *in vitro* amplification/selection of a similar prion isolate from brain and OM of FFI patients, which then resulted in a homogeneous pathological picture of the disease in both animals' groups (*Figure 4a*). The same brain homogenates were then treated with increasing concentrations of Gdn-HCl. Results of this analysis are shown in *Figure 4* and confirmed that prions found in the brains of FFI-BH_PMCA and FFI-OM_PMCA-inoculated animals were more stable than those found in the brain of FFI-BH-challenged mice. Notably, this difference was statistically significant, thus supporting the hypothesis that PMCA could have amplified a PrP$^{Sc}$ with distinctive biochemical and pathological features than that typically present in the FFI brain homogenate, finally leading to the development of different pathological changes in FFI-BH_PMCA and FFI-OM_PMCA groups of mice (also in this case, animal number #5 was excluded from the analysis) (*Figure 4b*). To deepen this aspect, we decided to investigate whether the PMCA itself might have modified the biochemical properties of the FFI-PrP$^{Sc}$ after amplification, prior to injection in BvPrP-Tg407 mice. To this aim, we have subjected FFI-BH and its product of amplification (FFI-BH_PMCA) to PK digestion and Gdn-HCl treatments. We have found that, after amplification, the PrP$^{Sc}$ did not change the PK-resistant properties (*Figure 4—figure*

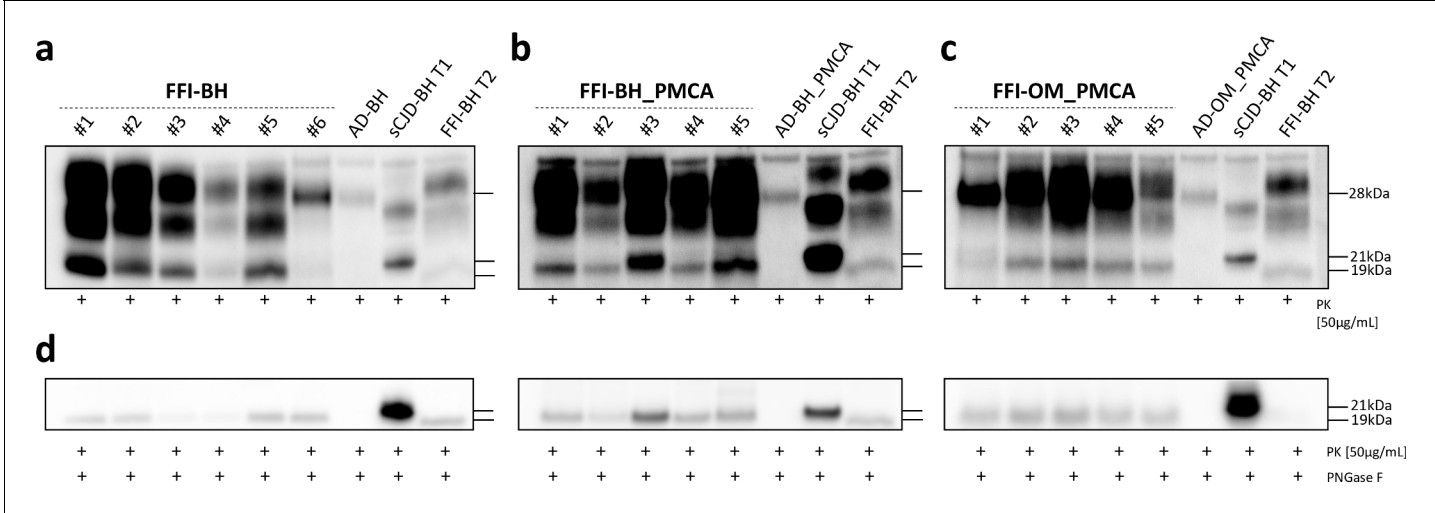

**Figure 3.** Western blot and PNGase F analysis of BvPrP-Tg407 brain homogenates. BvPrP-Tg407 mice inoculated with FFI-BH (**a**), FFI-BH_PMCA (**b**), and FFI-OM_PMCA (**c**) showed the presence of PrP^res while those inoculated with AD-BH, AD-BH_PMCA, and AD-OM_PMCA did not. All PrP^res were characterized by the predominance of the di-glycosylated band except for the animal number #5 of the group FFI-BH, which showed a PrP^res with the mono-glycosylated band predominant over the others. Samples were immunoblotted with anti-PrP 6D11 antibody. Similar findings have been observed using the Sha31 antibody (*Figure 3—figure supplement 1*). The brain homogenate of a FFI patient with PrP^res migrating at 19 kDa (FFI-BH T2) and the brain homogenate of a sCJD patient with PrP^res migrating at 21 kDa (sCJD-BH T1) were used as migration controls. PNGase F analyses showed that the un-glycosylated band migrated at 19 kDa in all cases, thus confirming the results obtained from animals with less clear migration pattern (e.g., animal number #3 of the FFI-BH_PMCA group and animal number #1 of the FFI-OM_PMCA group). Samples were immunoblotted with anti-PrP 6D11 antibody (**d**). FFI: Fatal Familial Insomnia; PMCA: Protein Misfolding Cyclic Amplification; OM: olfactory mucosa; BH: brain homogenate.

The online version of this article includes the following figure supplement(s) for figure 3:

**Figure supplement 1.** Western blot analysis of BvPrP-Tg407-inoculated mice.

supplement 2a) but became significantly more stable towards Gdn-HCl treatment (*Figure 4—figure supplement 2b*). Thus, we have observed that the PMCA has indeed slightly altered the biochemical properties of the original FFI-PrP^Sc, especially in terms of conformational stability.

We have then compared the biochemical properties of the PrP^Sc present in FFI-BH and the PrP^Sc generated in FFI-BH-injected mice. The results confirmed that the PrP^Sc present in FFI-BH was significantly less resistant to PK than that generated in BvPrP-Tg407-inoculated animals, thus suggesting that the original PrP^Sc properties were not retained upon animal transmission (*Figure 4—figure supplement 2a*). Then, we have also verified whether the biochemical properties of the PrP^Sc present in FFI-BH_PMCA changed after the inoculation in mice. In this case, we have found that PrP^Sc generated in mice was significantly more resistant to PK digestion and significantly less stable towards Gdn-HCl treatment (*Figure 4—figure supplement 2*). Therefore, also the PrP^Sc properties of FFI-BH_PMCA changed after animal transmission. Unfortunately, due to the lack of material we could not make similar comparisons between FFI-OM and FFI-OM_PMCA samples. However, it is conceivable that analogous events could have occurred even in this case.

## Neuropathological analysis of BvPrP-Tg407-inoculated mice

All groups of prion-inoculated animals showed mild spongiform changes mainly affecting the thalamus, the striatum, and to a lesser extent the frontal cortex. In particular, these alterations were slightly more pronounced in the brain of mice challenged with FFI-BH compared to that of the other groups. No vacuolation was observed in animals inoculated with AD-related samples (*Figure 5a*). Notably, the ventricles of most of the animals inoculated with FFI-BH were significantly enlarged (*Figure 5b*). Severe glial activation was observed in all groups of mice inoculated with FFI-related materials and mostly affected the thalamus. We did not find these alterations in the brain of all the other groups of mice inoculated with AD-related samples, which showed very faint glial activation, likely related to aging (*Figure 5b*).

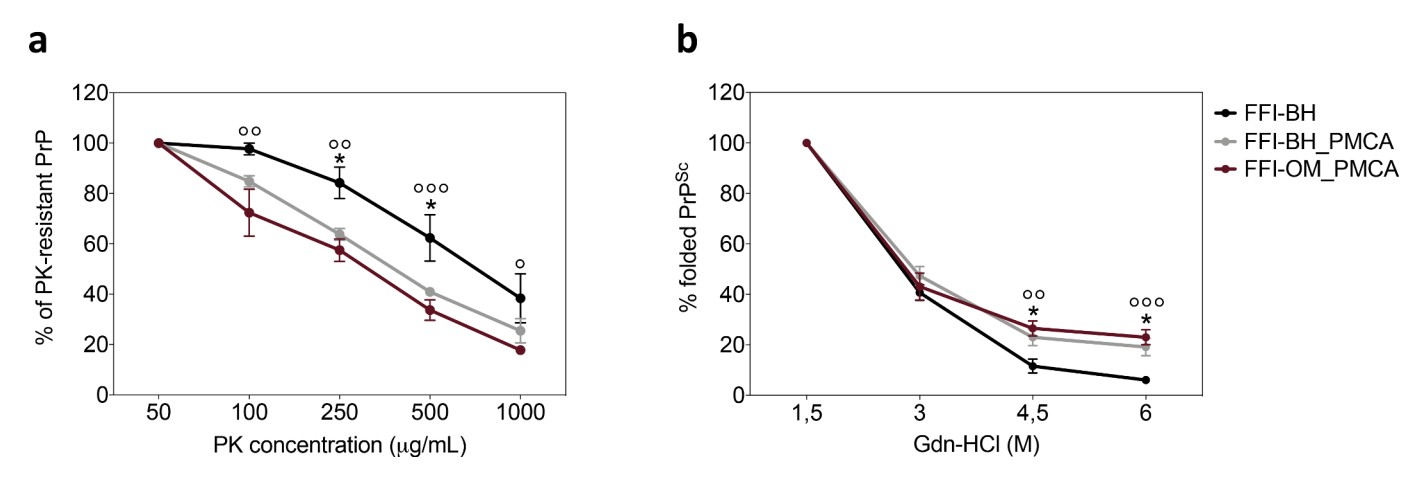

**Figure 4.** Biochemical characterization of prions generated in BvPrP-Tg407 mice. Proteinase K (PK) resistance assay (**a**) and conformational stability analysis (**b**) were performed to characterize the prions present in the brains of animals inoculated with FFI-BH, FFI-BH_PMCA, and FFI-OM_PMCA. These analyses revealed that prions found in the brain of FFI-BH-inoculated mice were more resistant to proteolytic digestion and less stable to guanidine hydrochloride (Gdn-HCl) treatment (black line) than those found in the brain of mice inoculated with FFI-BH_PMCA and FFI-OM_PMCA (gray and purple lines, respectively). In both cases, these differences were statistically significant (two-way ANOVA followed by Bonferroni post-tests; FFI-BH vs. FFI-BH_PMCA: *$p<0.05$; FFI-BH vs. FFI-OM_PMCA: ° $p<0.05$, °° $p<0.01$, °°° $p<0.001$; error bars: ± standard error of the mean [SEM]). In contrast, prions found in the brain of mice inoculated with FFI-BH_PMCA and FFI-OM_PMCA showed comparable PK-resistance profile and stability towards Gdn-HCl treatment. Separate PK and Gdn-HCl analysis were performed for animal number #5 that was excluded from the FFI-BH group (*Figure 4—figure supplement 1*). Analysis performed on FFI-BH sample and its PMCA product (FFI-BH_PMCA) showed that the amplification altered the PK and Gdn-HCl properties of PrP^Sc (*Figure 4—figure supplement 2*). FFI: Fatal Familial Insomnia; PMCA: Protein Misfolding Cyclic Amplification; OM: olfactory mucosa; BH: brain homogenate.
The online version of this article includes the following figure supplement(s) for figure 4:

**Figure supplement 1.** Proteinase K (PK)-resistance assay and conformational stability analysis of the prion present in the brains of animal number #5 excluded from the FFI-BH-injected group.

**Figure supplement 2.** Proteinase K (PK)-resistance assay and conformational stability analysis of the prions present in FFI-BH and its product of amplification (FFI-BH_PMCA) before and after injection in mice.

Animals inoculated with FFI-BH showed a synaptic and diffuse pattern of PrP^res deposition that mainly affected the thalamus, the striatum, and the deep layers of the frontal cortex. Plaque-like deposits (negative at ThS staining) were also found in the striatum and sometimes in the frontal cortex (*Figure 6*, *Figure 6—figure supplement 1*, *Figure 6—figure supplement 3*). Notably, animals #5 and #6 of this group did not show any PrP^res detectable by means of immunohistochemistry (although we could see a PrP^res signal by Wb) even in the thalamus and striatum where we observed a mild vacuolation and significant astroglial activation (*Figure 6—figure supplement 2*, *Figure 7*, *Figure 7—figure supplement 1*). These results were confirmed even after the treatment of the samples with lower concentrations of PK. Mice inoculated with FFI-BH_PMCA or FFI-OM_PMCA did not show any synaptic PrP^res deposition but only focal and plaque-like aggregates, negative at ThS staining (*Figure 6*, *Figure 6—figure supplement 3*) prevalently affecting thalamus, striatum, and cerebral cortex. These deposits were more abundant in animals inoculated with FFI-BH_PMCA compared to those inoculated with FFI-OM_PMCA (*Figure 6—figure supplement 1*). The possibility that this latter group of mice has developed a slightly different manifestation of the disease cannot be completely ruled out at this moment, and additional studies are needed to further investigate this aspect. As expected, no PrP^res immunoreactivity was found in the brain of mice inoculated with AD-BH, AD-BH_PMCA, or AD-OM_PMCA (*Figure 6*, *Figure 6—figure supplement 1*). Faint astroglial activation was observed in the striatum of these mice, but this might be associated with a normal aging process that is not an indicator of pathology (*Figure 7*, *Figure 7—figure supplement 1*).

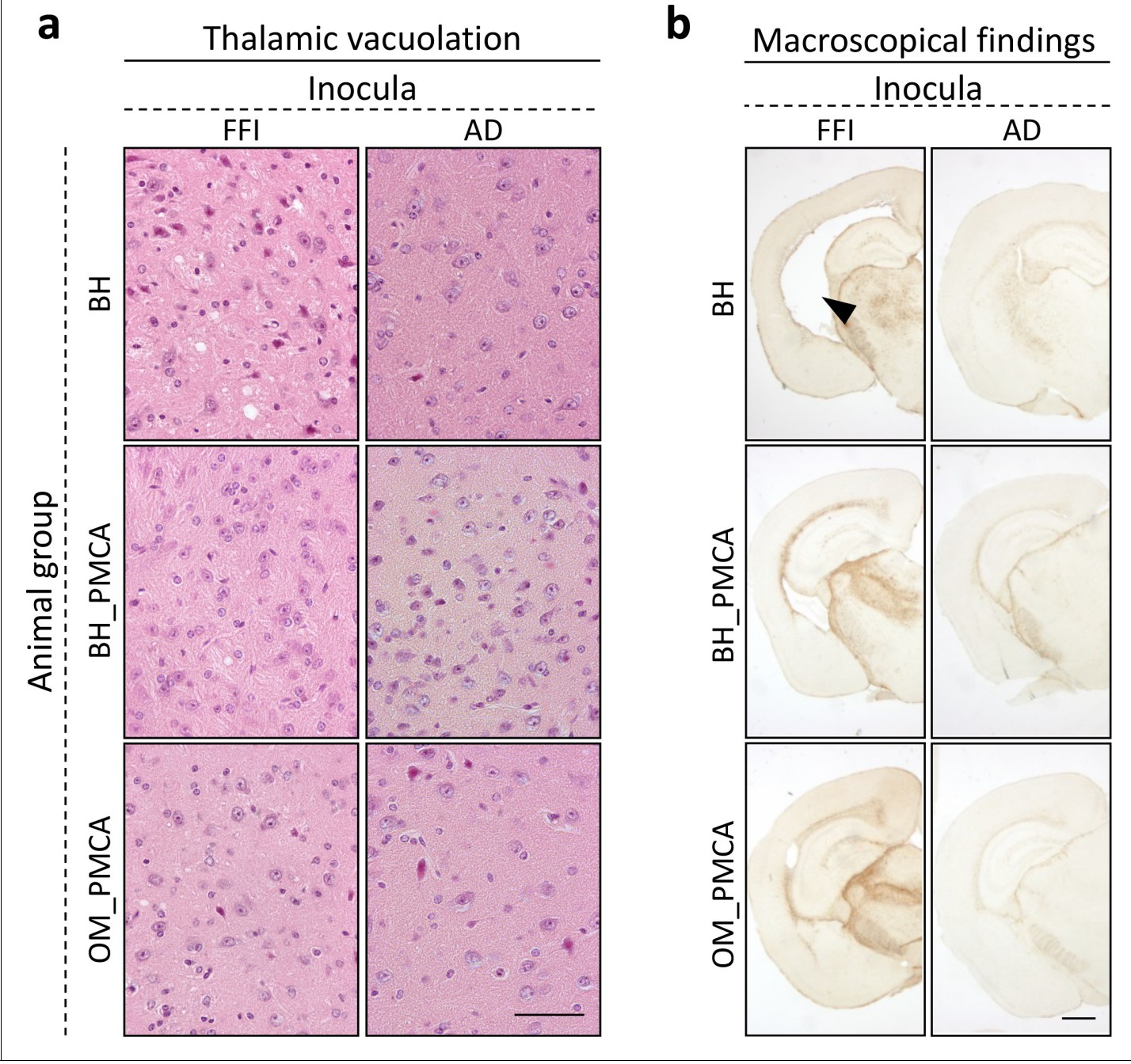

**Figure 5.** Histological findings. Faint vacuolation was observed in the thalamus of BvPrP-Tg407 mice inoculated with FFI-BH, FFI-BH_PMCA, and FFI-OM_PMCA. No spongiform changes were found in the brain of animals inoculated with AD-BH, AD-BH_PMCA, and AD-OM_PMCA. Sections were stained with hematoxylin and eosin. Scale bar: 50 µM (a). Mice inoculated with FFI-BH showed significant enlargement of the ventricles (see black arrow) that was not observed in the brain of all the other inoculated mice. Severe astroglial activations were observed in the thalamus of all prion-inoculated mice. Sections were immunostained with anti-glial fibrillary acidic protein antibody. Scale bar: 500 µM (b). FFI: Fatal Familial Insomnia; PMCA: Protein Misfolding Cyclic Amplification; OM: olfactory mucosa; BH: brain homogenate.

## RT-QuIC results

RT-QuIC analysis of BvPrP-Tg407 mice confirmed that the brains of animals inoculated with FFI-BH, FFI-BH_PMCA, and FFI-OM_PMCA were able to promote the aggregation of recBvPrP$_{90-231}$ with high efficiency. The aggregation kinetics were in general very rapid and the threshold of

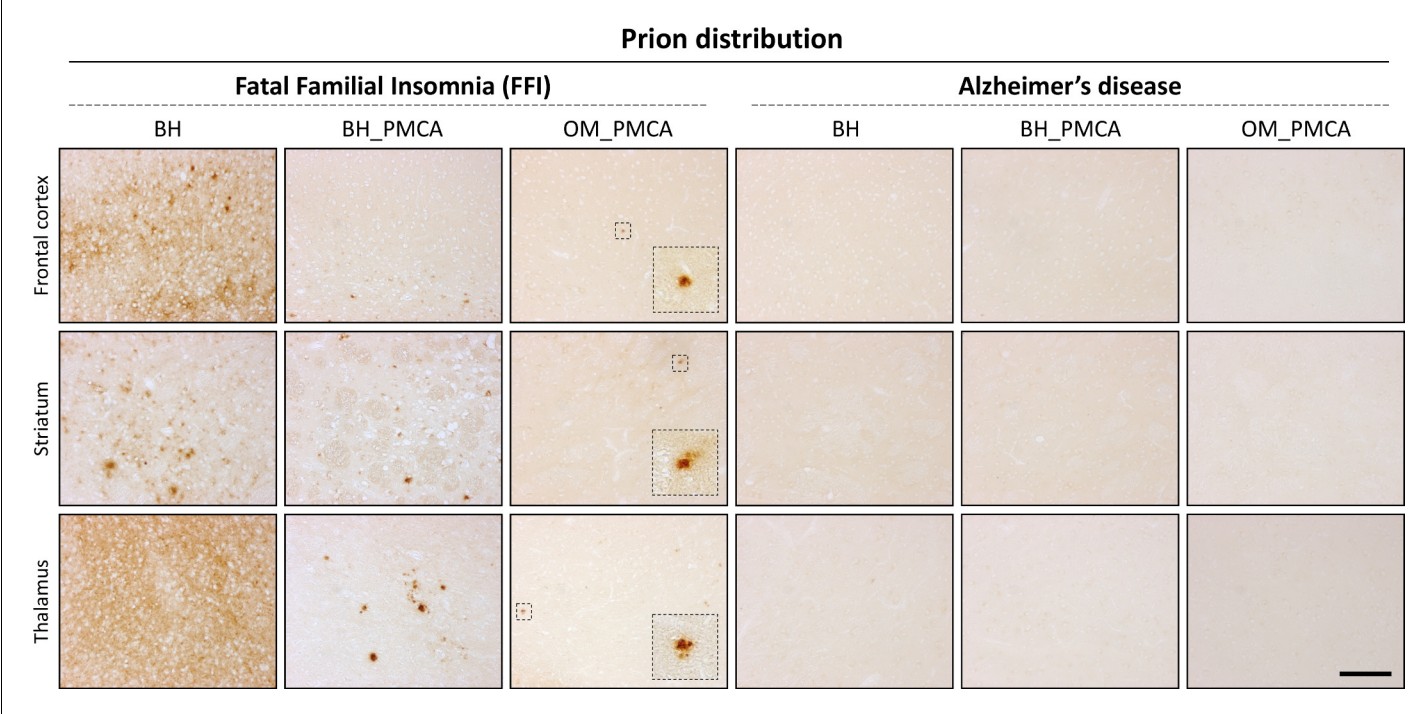

**Figure 6.** Prion distribution in the brain of BvPrP-Tg407-inoculated mice. BvPrP-Tg407 mice inoculated with FFI-BH showed synaptic distribution of PrPres with the presence of focal and plaque-like deposits mainly affecting the thalamus, striatum, and frontal cortex. Notably, animals #5 and #6 of this group did not show any PrPres (*Figure 6—figure supplement 2*). Animals inoculated with FFI-BH_PMCA showed only plaque-like deposits of PrPres mainly occurring in the thalamus and striatum. Similarly, but to a lesser extent, plaque-like deposits of PrPres were found in the thalamus, striatum, and frontal cortex of BvPrP-Tg407 mice inoculated with FFI-OM_PMCA (*Figure 6—figure supplement 1*). Focal deposits observed in the brain of FFI-BH, FFI-BH_PMCA, and FFI-OM_PMCA-injected mice were completely negative at ThS staining (*Figure 6—figure supplement 3*). No PrPres was found in the brain of mice inoculated with AD-BH, AD-BH_PMCA, and AD-OM_PMCA (*Figure 6—figure supplement 1*). Sections were immunostained with anti-PrP Saf34 antibody. Scale bar: 10 µm. FFI: Fatal Familial Insomnia; PMCA: Protein Misfolding Cyclic Amplification; OM: olfactory mucosa; BH: brain homogenate.

The online version of this article includes the following figure supplement(s) for figure 6:

**Figure supplement 1.** Schematic representation of prion distribution in the brain of BvPrP-Tg407-inoculated mice.
**Figure supplement 2.** Details of the prion distribution and astroglial activation found in the brain of mice numbers #5 and #6 inoculated with FFI-BH.
**Figure supplement 3.** Evaluation of the amyloid tinctorial properties of the PrPres deposits.

fluorescence was crossed by all samples before 10 hr, regardless of the inocula (*Figure 8*). From the analysis of the aggregation kinetics, we could not identify peculiar properties eventually useful to discriminate between the three different FFI inocula. Animals inoculated with AD-BH, AD-BH_PMCA, and AD-OM_PMCA induced recBvPrP$_{90-231}$ aggregation after 18 hr and were considered negative.

## Discussion

This is the second study showing that PrP$^{Sc}$ amplified by PMCA from peripheral tissues of patients with prion diseases is infectious when challenged in susceptible mice. In particular, our findings show that the PMCA-amplified products either generated from the brain or the OM of FFI patients are infectious when inoculated in transgenic mice expressing the BvPrP (BvPrP-Tg407). Remarkably, BvPrP-Tg407 mice do not develop spontaneous pathology (*Espinosa et al., 2016*). Regardless of the inoculum, all mice injected with PMCA products developed very similar neuropathological alterations characterized by focal and plaque-like deposits of PrP$^{res}$ mainly affecting the thalamus and the striatum. In contrast, animals inoculated with FFI brain homogenate (not subjected to PMCA) showed different histopathological features, for example, a stronger synaptic and diffuse PrP$^{res}$ immunoreactivity with focal and plaque-like deposits consistently affecting the thalamus, the striatum, and the deep layer of the cerebral cortex. Notably, the intensity of prion deposition observed

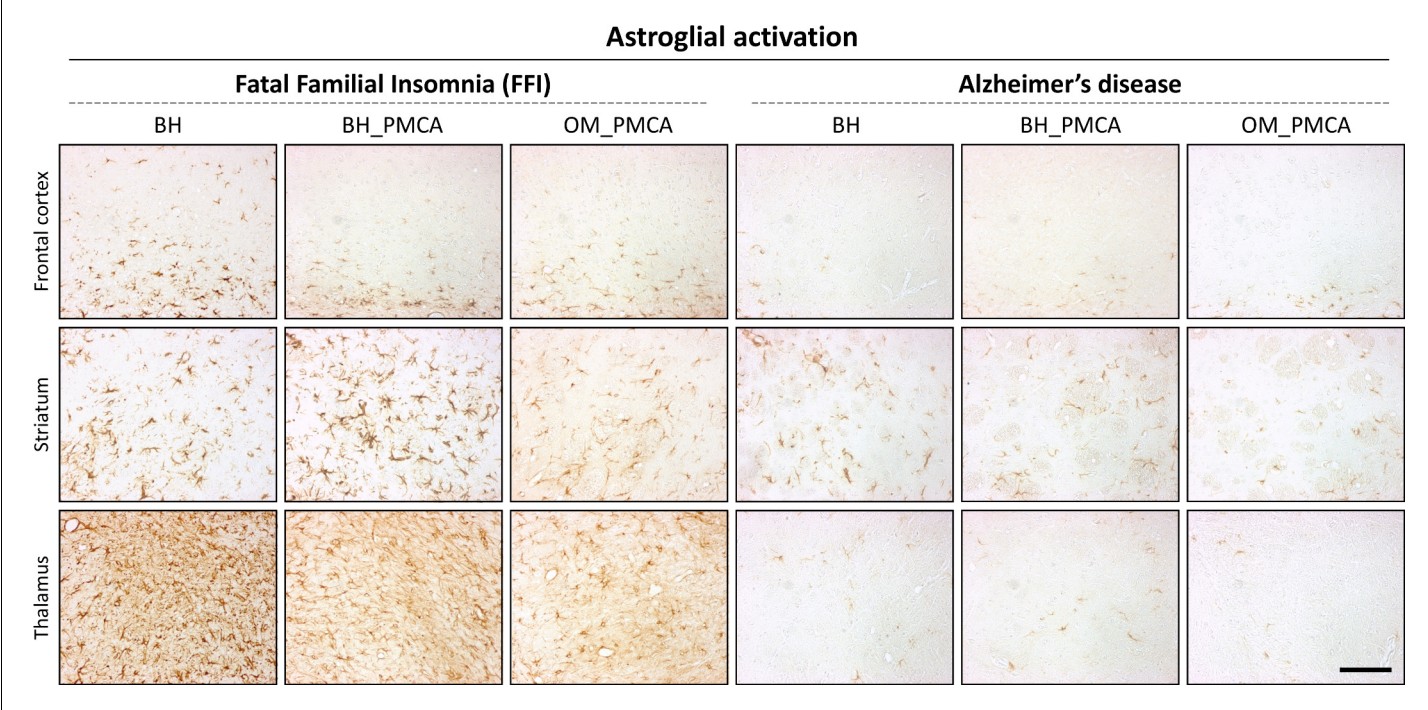

**Figure 7.** Astroglial activation in the brain of BvPrP-Tg407-inoculated mice. Animals inoculated with FFI-BH, FFI-BH_PMCA, and FFI-OM_PMCA showed severe glial activation in the thalamus and striatum. Moderate immunoreactivity was also found in the deep layer of the cerebral cortex of the animals inoculated with BH-FFI (*Figure 7—figure supplement 1*). Mild astroglial reactivity was observed in the striatum and thalamus of mice inoculated with AD-BH, AD-BH_PMCA, and AD-OM_PMCA. This activation is often detectable in healthy aged mice and is not an indicator of pathological processes (*Figure 7—figure supplement 1*). Sections were immunostained with anti-glial fibrillary acidic protein antibody. Scale bar: 10 μm. FFI: Fatal Familial Insomnia; PMCA: Protein Misfolding Cyclic Amplification; OM: olfactory mucosa; BH: brain homogenate.

The online version of this article includes the following figure supplement(s) for figure 7:

**Figure supplement 1.** Schematic representation of the astroglial activation in the brain of BvPrP-Tg407-inoculated mice.

in all mice was paralleled by a proportional glial activation. Unfortunately, we did not have the possibility to inject the raw FFI-OM because the original sample underwent several analyses (including extensive PMCA and RT-QuIC investigations, as reported in the work of *Redaelli et al., 2017*) and we did not have enough material for the inoculation. An additional OM sampling (finalized to animals' injection) was not possible because the patient died soon after the first OM collection. However, a very recent study (*Raymond et al., 2020*) showed that OM of patients with sporadic CJD are able to induce prion disease when challenged in transgenic mice overexpressing the human PrP with methionine at codon 129 (Tg66). In this study, only a small percentage of animals developed the disease, thus suggesting that the infectivity in OM is very low.

The use of BvPrP-Tg407 mice was dictated by the fact that the PrP$^{Sc}$ present in OM of the FFI patient was efficiently amplified by using the brain of bank vole (Bv109M) as a PMCA reaction substrate (*Redaelli et al., 2017*). Therefore, to avoid additional passages of species, we decided to inject BvPrP-Tg407 mice while being conscious that the original properties of the FFI prions could have been altered. However, after injection, this model has generated a bank vole-adapted FFI prion isolate that, to some extent, retained few features of the original human FFI strain.

We have attempted to amplify the FFI-PrP$^{Sc}$ from OM by using the brain of mice expressing the human PrP with methionine at codon 129 of *PRNP* (HuPrP129MM) as a PMCA substrate, but we were not successful. Using this model, we have shown that the PrP$^{Sc}$ amplified from the urine of patients with the new variant CJD (vCJD) was able to maintain the same biochemical and neuropathological features of the vCJD-PrP$^{Sc}$ present in the brain (*Cali et al., 2019*). However, in the present work, we were required to use the bank vole PrP$^{C}$ as an efficient acceptor of the FFI prions and our results need to take into consideration these experimental constraints.

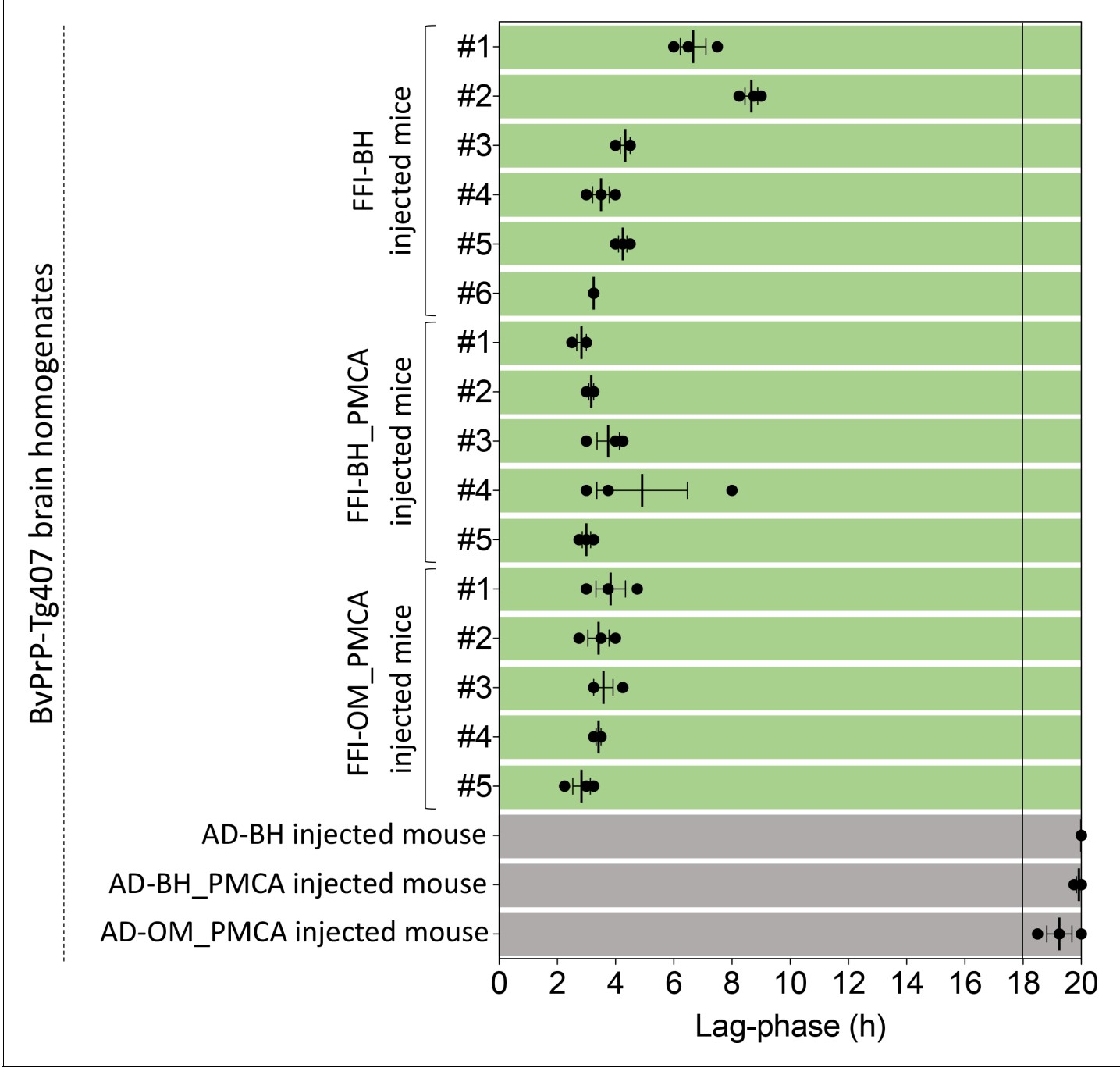

**Figure 8.** Real-Time Quaking-Induced Conversion (RT-QuIC) results of BvPrP-Tg407 brain homogenates. All brain homogenates of the animals inoculated with FFI-BH, FFI-BH_PMCA, and FFI-OM_PMCA induced a rapid aggregation of the recBvPrP$_{90-231}$ used as a RT-QuIC reaction substrate. Aggregation of the substrate was observed also in the presence of brain homogenates of mice inoculated with AD-BH, AD-BH_PMCA, and AD-OM_PMCA but occurred after 18 hr and were considered negative. Error bars:± standard error of the mean (SEM). FFI: Fatal Familial Insomnia; PMCA: Protein Misfolding Cyclic Amplification; OM: olfactory mucosa; BH: brain homogenate.

The first important finding of this study was that BvPrP-Tg407 mice were highly susceptible to human FFI prions (100% attack rate). It is known that the transmission of prions between species is an inefficient process because of the species barrier that is mainly dictated by the amino acid differences between the PrP$^{Sc}$ of the donor and the PrP$^{C}$ of the acceptor species, and the conformational properties of PrP$^{Sc}$ aggregates (*Hagiwara et al., 2013*). It is known that the BvPrP can be converted

into PrP$^{Sc}$ by a number of human and animal prion strains with high efficiency (*Watts et al., 2014*; *Nonno et al., 2006*; *Agrimi et al., 2008*). There is only one work showing that FFI prions are efficiently transmissible to knock-in mice expressing BvPrP with methionine at codon 109 of *Prnp* (*Takeuchi et al., 2019*). Our work confirmed that also our model is susceptible to FFI. In particular, animals inoculated with FFI-BH showed mild spongiform changes and glial activation mainly localized in the thalamus, striatum, and deep layer of the frontal cortex. They accumulated PrP$^{res}$ that was characterized by the typical FFI glycoform profile: predominance of the di-glycosylated PrP band and migration of the un-glycosylated band at 19 kDa. However, several lines of evidence indicate that the BvPrP is prone to convert into PrP$^{Sc}$ while retaining these biochemical features, regardless of the inocula (*Espinosa et al., 2016*). For instance, upon inoculation with different human and animal prion strains, transgenic mice expressing the BvPrP with methionine or isoleucine at codon 109 (TgM109 or TgI109, respectively) developed prion pathology and all generated PrP$^{res}$ were characterized by a predominance of the di-glycosylated band, even in those challenged with human or animal prion strains with the mono-glycosylated predominant PrP band (e.g., sporadic CJD or Rocky Mountain Laboratory prion strain (RML)) (*Watts et al., 2014*). This has been observed also in transgenic mice expressing BvPrP with isoleucine at codon 109 and the D178 mutation that spontaneously developed prion diseases with the presence of a PrP$^{res}$ with a predominant di-glycosylated band (*Watts et al., 2016*). Many other studies showed that this phenomenon occurred also in other transgenic mouse models inoculated with FFI, thus suggesting that, regardless of the recipient mice, some biochemical features typical of the FFI are reproduced after inoculation (*Telling et al., 1996*; *Takeuchi et al., 2019*).

For this reason, although the FFI glycoform profile was maintained upon transmission in BvPrP-Tg407 mice, we need to consider that this finding might be related to some intrinsically features of the animal model used. Another alteration that we have noticed in the brain of many FFI-BH-inoculated mice, as previously published (*Jackson et al., 2009*), was a significant enlargement of the ventricular spaces. This change was not found in the brain of mice inoculated with FFI-BH_PMCA and FFI-OM_PMCA. Surprisingly, in one animal of the FFI-BH-inoculated group we have identified a PrP$^{res}$ form characterized by the predominance of the mono-glycosylated band with the un-glycosylated one always migrating at 19 kDa. In addition, compared to all the other prions of this group, it differentially behaved to PK and Gdn-HCl treatments (*Figure 4—figure supplement 1*) and PrP$^{Sc}$ was not detectable by immunohistochemistry, even by treating the samples with lower concentrations of PK. Finally, this animal did not show the typical ventricles enlargement observed in the brains of the other mice of this group. Therefore, we decided to exclude this animal from the rest of the group in consideration of the fact that it might have generated a distinct prion isolate. There is a compelling evidence that some human prion strains originally believed to be 'pure' are instead composed by a mixture of PrP$^{Sc}$ variants (*Cassard et al., 2020*; *Kobayashi et al., 2011*; *Bishop et al., 2010*; *Morales et al., 2016*). Interestingly, a very recent publication of the group of Kitamoto Tetsuyuki showed that even the FFI$^{D178N-129MM}$ strain might not be pure and that at least two distinct prion strains can be responsible for the disease, one associated with the typical clinical presentation of the disease and the other associated with an atypical phenotype (*Takeuchi et al., 2019*). It is therefore conceivable that the mono-glycosylated PrP$^{res}$ observed in one of the animals inoculated with FFI-BH might have emerged either because the FFI propagates as a mixture of prions (as also shown for other sporadic forms of prion diseases) or as a random event due to specific limitations imposed by the animal model used. Indeed, these mice have never been challenged with FFI before and seem to be potentially permissive to underrepresented prion strains eventually coexisting in the same isolate.

Prior to injection in BvPrP-Tg407 mice, we wanted to assess whether and to what extent the PMCA analysis could have modified the original features of PrP$^{Sc}$ present in FFI-BH, eventually generating a different infectious prion isolate. To this aim, we have performed PK and Gdn-HCl characterization of PrP$^{Sc}$ present in FFI-BH and FFI-BH_PMCA and found that after PMCA the biochemical properties of PrP$^{Sc}$ have slightly changed. Unfortunately, due to lack (raw OM) or insufficient material (FFI-OM_PMCA that was thoroughly inoculated in mice), we could not evaluate whether the amplification of FFI-OM might have generated a PrP$^{Sc}$ with different biochemical properties. We have then evaluated the biochemical and neuropathological alterations developed in the brain of all injected mice, with a special focus on those challenged with FFI-BH, FFI-BH_PMCA, and FFI-OM_PMCA. Notably, mice inoculated with FFI-BH and FFI-BH_PMCA developed distinct

pathologies, while those inoculated with FFI-OM_PMCA showed alterations analogous to those found in FFI-BH_PMCA-challenged animals, although less pronounced. For this reason, the PMCA might have produced prion isolates with similar infectious properties, regardless of the tissue used for the amplification (BH vs. OM). This evidence prompted us to speculate that prions present in the brain and OM could be similar, thus explaining the pathological analogies observed in mice challenged with PMCA-generated products. However, it should be considered that other factors (e.g., genetic or environmental) of the BvPrP-Tg407 recipient mice might have further controlled the formation of these prions. What is certain, in any case, is that these PMCA-produced prions are not the results of a *de novo* generation since we did not observe any PrP$^{res}$ in amplified brain or OM samples collected from the AD patient that were subjected to the same amplification procedures. Overall, these products did not induce pathology when challenged in BvPrP-Tg407 mice. To confirm the absence of prions (even trace amounts) in the brain of these animals, we have performed RT-QuIC experiments that showed the lack of seeding activity in all AD-related samples. As expected, strong RT-QuIC responses were obtained from the analyses of mice inoculated with FFI-related material.

Taken together these results showed that the FFI brain homogenate was able to induce pathology in BvPrP-Tg407 mouse model and promoted the generation of two distinct prion isolates, one more represented that was characterized by the prevalence of the di-glycosylated band (83%) and another less represented characterized by the prevalence of the mono-glycosylated band (17%). Both isolates showed distinct features and support the fact that FFI might not be a pure strain, even among FFI$^{D178N-129MM}$ subjects. As previously mentioned, all PMCA products were infectious and elicited analogous biochemical and neuropathological changes when challenged in BvPrP-Tg407 mice, probably due to prior selective effects of the PMCA on prions present in the brain or OM of the FFI patient. To better investigate this aspect, we have compared the biochemical properties of PrP$^{Sc}$ present in FFI-BH_PMCA (prior to passaging in mice) with those of the PrP$^{Sc}$ generated in the brain of FFI-BH_PMCA-injected mice. We have observed that the inoculation generated a PrP$^{Sc}$ significantly more resistant to PK digestion and less stable against Gdn-HCl compared to that of the inoculum. Thus, it might be hypothesized that other than a possible PMCA selection of prions present in the brain or OM of the FFI patients, additional processes of adaptation might have occurred in the brain of inoculated mice.

In contrast, animals inoculated with AD-BH, AD-BH_PMCA, and AD-OM_PMCA did not show any prion-related pathology.

From this study, we can conclude that BvPrP-Tg407 mice are susceptible to both the FFI prions and their bank vole-adapted PMCA-generated products. We were intrigued by the fact that mice inoculated with FFI-OM_PMCA developed less PrP$^{res}$ deposits compared to those inoculated with FFI-BH_PMCA. However, by using the methods generally employed to study prion strains (e.g., PK, Gdn-HCl, PNGase F assays), we did not detect other differences supporting that this group of mice might have developed a distinct prion isolate from that observed in FFI-BH_PMCA-inoculated animals. Additional and more advanced studies are needed to further investigate this aspect. Moreover, we have noticed that animal number #1 of the FFI-OM_PMCA group showed two distinct unglycosylated PrP$^{res}$ fragments, thus further supporting the possibility that PMCA might have generated a multitude of unstable prion isolates capable of inducing slightly distinct alterations even within the same group of animals. This finding was detectable only with the use of the N-terminal 6D11 antibody and was not observed with the use of the C-terminal Sha31 antibody.

In summary, our injected mice developed prion pathologies that recapitulate some of the key clinical hallmarks of FFI (e.g., vacuolation, glial activation, and PrP$^{res}$ deposition mainly affecting the thalamus) and, although with some limitations, this model can be used to (1) deepen some of the pathological events associated with FFI, (2) uncover new factors eventually involved in its phenotypic heterogeneity, and (3) evaluate the effects of therapeutic compounds to interfere with the progression of the disease in a more phenotypic-specific way. The recent discovery that FFI patients with the same genotype might have distinct clinic-pathological presentations of the disease suggests that still unidentified factors (including the presence of distinct PrP$^{Sc}$ strains), other than the genetic background, may be responsible for the pathogenesis of the disease. This evidence clearly indicates that the mutation can trigger the onset of the disease but many other factors modulate the pathological process, which results in a heterogeneous presentation of the disease. The possibility of detecting PrP$^{Sc}$ in the OM can be exploited for basic research experiments, for therapeutic studies but also for monitoring disease progression, especially in patients eventually enrolled in clinical trials. This study

shows that OM contains traces of FFI-PrP$^{Sc}$ that can be efficiently amplified by PMCA using the brain of bank vole as substrate and that the amplified products are infectious when challenged in mice that express the bank vole PrP$^C$ (BvPrP-Tg407).

The fact that the PMCA products are as infectious as the FFI brain homogenate raises major biosafety concerns that require the adoption of specific precautions when manipulating amplified material (*Bistaffa et al., 2017*). For diagnostic purpose, a safer alternative is represented by the RT-QuIC assay that, according to the latest evidence, was found not to replicate biohazardous prions *in vitro* (*Raymond et al., 2020*).

Although with some experimental constraints, we showed that the OM of FFI patients contains potentially infectious prions. All inoculated animals developed pathology in the thalamus, which is also the area typically affected in the brain of patients with FFI. Moreover, the glycoform ratio of the PrP$^{res}$ was similar to that found in the brain of diseased patients. Therefore, some of the typical features of the FFI-PrP$^{Sc}$ have been carefully reproduced in this animal model, even in samples that underwent PMCA treatment before the injection. We are aware of the fact that the experiments have been performed using the bank vole PrP$^C$ as acceptor of FFI prions, either *in vitro* (PMCA with Bv109M substrate) or *in vivo* (BvPrP-Tg407). Hence, some important FFI-PrP$^{Sc}$ features might have been lost due to the passage of the species barrier, as already shown by several experimental transmission studies described in the literature (*Bartz et al., 2000*; *Peretz et al., 2002*; *Bruce et al., 1994*). Additional studies with more suitable *in vitro* and *in vivo* models are still needed to investigate whether the properties of the FFI-PrP$^{Sc}$ present in the OM and brain are similar or whether they are influenced by the environment of the tissues. Having the possibility to collect (with non-invasive procedures) and amplify PrP$^{Sc}$ from the OM of FFI patients and test its infectious properties in mice will consent to deepen the pathological mechanisms associated with individual disease phenotype, identify the factors (other than the mutation and the polymorphism at codon 129) eventually involved in the phenotypic heterogeneity of the disease, and uncover new potential targets for the development of individual therapies capable of blocking the progression of this devastating condition. An increasing number of studies are being published to support the important role that the OM might have for research and diagnosis purposes in the field of prion and other neurodegenerative diseases (*Redaelli et al., 2017*; *Raymond et al., 2020*; *Orrú et al., 2014*; *De Luca et al., 2019*).

## Acknowledgements

We thank Associazione Italiana Encefalopatie da Prioni (AIEnP). *Funding*: This work was supported in part by the Italian Ministry of Health (GR-2013-02355724 and Ricerca Corrente), MJFF, ALZ, Alzheimer's Research UK and the Weston Brain Institute (BAND2015), and Euronanomed III (SPEEDY) to FM; by the Spanish Ministerio de Economía y Competitividad (grant AGL2016-78054-R [AEI/FEDER, UE]) to JMT and JCE; AM-M was supported by a fellowship from the INIA (FPI-SGIT-2015-02). The funders had no role in study design, data collection and interpretation, or the decision to submit the work for publication.

## Additional information

### Funding

| Funder | Grant reference number | Author |
|---|---|---|
| Ministero della Salute | GR-2013-02355724 | Fabio Moda |
| Ministero della Salute | Ricerca Corrente | Fabio Moda |
| MJFF, ALZ, Alzheimer's Research UK and the Weston Brain Institute | BAND2015 | Fabio Moda |
| Euronanomed III | SPEEDY | Fabio Moda |
| Ministerio de Economía y Competitividad | AGL2016-78054-R (AEI/FEDER,UE) | Juan Maria Torres |
| INIA | Fellowship FPI-SGIT-2015-02 | Alba Marín Moreno |

The funders had no role in study design, data collection and interpretation, or the decision to submit the work for publication.

## Author contributions
Edoardo Bistaffa, Data curation, Formal analysis, Investigation, Methodology, Writing - original draft, Writing - review and editing; Alba Marín-Moreno, Data curation, Investigation, Methodology, Writing - original draft; Juan Carlos Espinosa, Federico Angelo Cazzaniga, Data curation, Investigation, Methodology; Chiara Maria Giulia De Luca, Sara Maria Portaleone, Luigi Celauro, Investigation, Methodology; Giuseppe Legname, Giorgio Giaccone, Supervision, Writing - original draft, Writing - review and editing; Juan Maria Torres, Conceptualization, Supervision, Funding acquisition, Writing - original draft, Writing - review and editing; Fabio Moda, Conceptualization, Data curation, Formal analysis, Supervision, Funding acquisition, Writing - original draft, Project administration, Writing - review and editing

## Author ORCIDs
Edoardo Bistaffa (iD) https://orcid.org/0000-0001-7441-153X
Juan Carlos Espinosa (iD) http://orcid.org/0000-0002-6719-9902
Juan Maria Torres (iD) http://orcid.org/0000-0003-0443-9232
Fabio Moda (iD) https://orcid.org/0000-0002-2820-9880

## Ethics
Human subjects: Written informed consent for participation in research and all procedures for sample collection and experimental studies were in accordance with the 1964 Declaration of Helsinki and its later amendments and were approved by the Ethical Committee of "Fondazione IRCCS Istituto Neurologico Carlo Besta" (Milan, Italy).
Animal experimentation: We conducted animal experiments in accordance with the Code for Methods and Welfare Considerations in Behavioural Research with Animals (Directive 2010/63/EU) and made every effort to minimize suffering. Experiments developed in Centro de Investigación en Sanidad Animal-Instituto Nacional de Investigación y Tecnología Agraria y Alimentaria (Madrid, Spain) were evaluated by the Committee on the Ethics of Animal Experiments of the Instituto Nacional de Investigación y Tecnología Agraria y Alimentaria and approved by the General Directorate of the Madrid Community Government (permit no. PROEX 263-15).

## Decision letter and Author response
Decision letter https://doi.org/10.7554/eLife.65311.sa1
Author response https://doi.org/10.7554/eLife.65311.sa2

# Additional files

## Supplementary files
• Transparent reporting form

## Data availability
All data generated or analyzed during this study are included in the manuscript.

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
