## [Decision Letter]

**Acceptance summary:**

This manuscript reports protein misfolding cyclic amplification (PMCA) of PrPSc from the olfactory mucosa of fatal familial insomnia (FFI) patients, successful transmission, and strain adaptation when passaging into a new host. This work demonstrates the infectivity and adaptability of PMCA-amplified PrPSc from FFI patients. The finding that olfactory mucosa of FFI carrying infectious risk is novel and important, with implications for nasal swabs taken from large populations of patients (for example, in a coronavirus pandemic). The work also raises new questions about how infectious particles reach reach the olfactory mucosa in FFI, which tends to be more anatomically restricted than other prion diseases.

**Decision letter after peer review:**

Thank you for submitting your article "PMCA generated prions from the olfactory mucosa of patients with Fatal Familial Insomnia cause prion disease in mice" for consideration by *eLife*. Your article has been reviewed by 3 peer reviewers, and the evaluation has been overseen by a Reviewing Editor and Mone Zaidi as the Senior Editor. The following individual involved in review of your submission has agreed to reveal their identity: Shu Chen (Reviewer #1).

Essential Revisions:

1. Based on the different sensitivities to proteolysis and guanidine denaturation among the brains of those mice inoculated with PMCA generated PrPSc versus those inoculated with FFI brain homogenates, the authors conclude that a bank vole adapted strain of FFI may have been generated. This conclusion could be strengthened by performing these same assays on the PMCA generated PrPSc prior to inoculation. This would demonstrate whether these biochemical differences existed prior to passaging in the BvPrP mice. In addition, it would be beneficial to assess the properties of the original FFI brain homogenate and how it may have changed when performing PMCA or passaging experiments.

2. The PrPSc deposition and glial pathology in Figures 6 and 7, respectively, are very difficult to see and compare among cohorts. Images with higher magnification or insets would be helpful.

3. The authors include an observation in the discussion highlighting that one of the FFI-OM_PMCA animals had 2 distinct un-glycosylate PrP bands. This was not mentioned in anywhere in the Results section. Please discuss this observation in the appropriate Results section.

4. Some experimental details should be better described. These include the genotypes of PRNP of the two FFI patients, and whether PMCA products used for inoculation were derived from one or both patients, and which rounds of the PMCA cycles were used in this study. Also needed are the stains and antibodies used in the legends of relevant figures.

5. The discussion may also include the biosafety precautions for PMCA products, as well as recent studies of RT-QuIC products which appears to be a safer alternative for diagnostic purpose in light of a recent negative finding in OM of CJD.

Reviewer #1:

This work provides a novel insight into risk of prion transmission associated with olfactory mucosa (nasal brushings) from patients with an inherited prion disease called fatal familial insomnia (FFI). The culprit of prion disease and its transmission is a misfolded form prion protein called PrPSc. Because OM, and likely other peripheral tissues, presumably contain only traces amounts of PrPSc and cannot be detected by conventional methods, new technology for amplified production and detection of PrPSc has emerged recently including PMCA (Protein Misfolding Cyclic Amplification) or RT-QuIC (real-time quaking induced conversion) assays. The authors previously succeeded in detecting traces of PrPSc in OM of FFI patients using PMCA that converts normal cellular PrP (PrPC) into PrPSc-like aggregates, allowing for a diagnostic assay. In this study, the authors attempt to assess whether the in vitro generated PMCA products, the surrogate of PrPSc in original OM, are infectious and able to transmit prion disease in an animal model. Their findings may have important implications regarding the potential risk of prion transmission through iatrogenic exposures of patient's peripheral specimens. Overall, the conclusion of this paper that OM of FFI patients is potentially infectious are mostly supported by data as presented.

Strengths:

1. The study design contained test groups and controls appropriate for hypothesis testing. The test groups included brain homogenate of FFI (FFI_BH), its PMCA products (FFI_BH_PMCA) and PMCA products from OM of FFI (FFI_OM_PMCA). The control group included corresponding materials from Alzheimer's disease (AD), a non-prion neurodegenerative disease. Thus, a total of six groups were tested by inoculation of these specimens into brains of an animal model susceptible to human prions, a transgenic mice expressing bank vole PrP (BvPrP-Tg407). As their data shown, inoculation of FFI-BH, FFI_BH_PMCA, and FFI_OM_PMCA all caused prion pathology in brains of BvPrP-Tg407 mice, while no prion transmission occurred by inoculation of AD brain homogenate (AD_BH) or PMCA products from BH and OM of AD (AD_BH_PMCA and AD_OM_PMCA). This set of experiments allowed the authors to conclude that a) PMCA products of OM from FFI were infectious; the inoculated animals all succumbed to prion disease (100% attack rate), similar to those inoculated with FFI_BH_PMCA or FFI_BH; b) No prion transmission occurred following inoculation of PMCA products from AD, excluding the possibility that PMCA generated de novo prion from unaffected brains.

2. The use of BvPrP-Tg407 mice as a highly susceptible recipient for prion transmission from FFI allowed robust analysis of distinct prion pathology and prion strain diversity. For example, inoculation of FFI_BH caused spongiform degeneration, enlarged ventricles, and synaptic deposition of PrPSc mainly in thalamus in BvPrP-Tg407 mice, thus recapitulating neuropathological features of human FFI. The PMCA products (FFI_BH_PMCA and FFI_OM_PMCA) seemed to cause a slightly different pathological features with less spongiform formation and only focal PrPSc deposition, also in thalamus. The authors attributed such difference to the possibility that PMCA products may represent bank vole adapted prions. This is reasonable in view of the previous transmission studies. Interestingly, one out of six animals inoculated with FFI_BH produced PrPSc that had different biochemically properties from the reset of the inoculated animals, as well as a different neuropathological profile. This raised the possibility that some FFI patients may harbor more than one prion strains, again consistent with the findings from other studies.

3. The paper utilized many different experimental approaches in characterizing PrPSc-containing materials. These included Western blot analysis of protease-resistant PrP (PrPres) before and after inoculation, detailed neuropathological proofing of inoculated animal brains, and secondary measures by RT-QuIC assay.

4. In the context of the literature, this study represent the first study of prion transmissibility of PMCA products from FFI, a human prion disease that is known to have lower levels of PrPSc in select regions of brains such as thalamus. In CJD that contains high levels and more broad distribution of PrPSc in affected brains, PMCA products amplified from urine of variant CJD was shown to be infectious. The raw OM of sporadic CJD was also recently reported to be transmissible albeit with low rate of infectivity using a different animal recipient. Therefore, the present study strengthens the evidence that despite likely low infectivity in peripheral tissues such as OM, the potential risk of prion transmission should not be ignored especially during tissue transplantation and surgical interventions.

This is an interesting paper that provides a novel insight into the transmissibility of peripheral tissues from a routinely accessible tissue including olfactory mucosa from patients with FFI.

Several points of clarification may strengthen the science of this paper.

1. Some experimental details should be better described. These include the genotypes of PRNP of the two FFI patients, and whether PMCA products used for inoculation were derived from one or both patients, and which rounds of the PMCA cycles were used in this study. Also needed are the stains and antibodies used in the legends of relevant figures.

2. The discussion may also include the biosafety precautions for PMCA products, as well as recent studies of RT-QuIC products which appears to be a safer alternative for diagnostic purpose in light of a recent negative finding in OM of CJD.

Reviewer #2:

In this study, Bistaffa et al. investigate the transmissibility of PMCA generated PrPSc from the olfactory mucosa of Fatal Familial Insomnia (FFI) patients. This is an extension of the author's previous work demonstrating the amplification of PrPSc from the olfactory mucosa (OM) of FFI patients through the use of PMCA. In this manuscript, they inoculate the PMCA generated PrPSc derived from the OM into the brains of knock-in mice expressing the bank vole PrP (BvPrP). They test this preparation against several controls including PMCA generated material from the brains of FFI patients, PMCA generated material from the OM and brains of Alzheimer's disease (AD) patients, or brain homogenate from FFI and AD patients. The authors demonstrate that the PMCA amplified PrPSc from both the brains and OM of FFI patients was capable of inducing disease in the BvPrP expressing mouse at a similar time as those mice injected with FFI brain homogenate. Homogenates of AD derived tissue did not cause disease. The brains of the diseased mice, independent of the inocula, only showed mild spongiform pathology. PrPSc deposition, however, was more robust in the FFI brain homogenate inoculated animals than in those inoculated with PMCA generated PrPSc. Furthermore, the brains from diseased animals inoculated with PMCA generated prions were observed to be less resistant to proteolysis and more stable to guanidine denaturation than the brains of those animals inoculated with FFI brain homogenate. The authors conclude that passaging FFI through bank vole PrP, both through PMCA and in vivo may have generated a bank vole-adapted strain of FFI.

The conclusions of the paper are supported by the data, but there are additional controls and information that could be included that would strengthen the manuscript.

1. Based on the different sensitivities to proteolysis and guanidine denaturation among the brains of those mice inoculated with PMCA generated PrPSc versus those inoculated with FFI brain homogenates, the authors conclude that a bank vole adapted strain of FFI may have been generated. This conclusion could be strengthened by performing these same assays on the PMCA generated PrPSc prior to inoculation. This would demonstrate whether these biochemical differences existed prior to passaging in the BvPrP mice. In addition, it would be beneficial to assess the properties of the original FFI brain homogenate and how it may have changed when performing PMCA or passaging experiments.

2. The PrPSc deposition and glial pathology in Figures 6 and 7, respectively, are very difficult to see and compare among cohorts. Images with higher magnification or insets would be helpful.

3. The authors include an observation in the discussion highlighting that one of the FFI-OM_PMCA animals had 2 distinct un-glycosylate PrP bands. This was not mentioned in anywhere in the Results section. Please discuss this observation in the appropriate Results section.

Reviewer #3:

The experiments are well-designed, logically organized, and convincingly show that PMCA successfully amplified FFI PrPSc. The amplified PrPSc was infectious to mice expressing BV PrPC, retaining some features of FFI PrPSc, but clearly selecting for a PrPSc conformer with differing biochemical properties than the original FFI prion, and thereby inducing a somewhat different pathologic profile. The experimental models are appropriate.

The main issue lies in the novelty of the work and the scientific advance, since FFI infectivity in mice expressing bank vole PrPC has been shown, and FFI PrPSc has been previously amplified by PMCA.

---

## [Author Response]

Reviewer #1:[…] Several points of clarification may strengthen the science of this paper.1. Some experimental details should be better described. These include the genotypes of PRNP of the two FFI patients, and whether PMCA products used for inoculation were derived from one or both patients, and which rounds of the PMCA cycles were used in this study. Also needed are the stains and antibodies used in the legends of relevant figures.

As suggested by the reviewer, we have added some experimental details. In particular, we have:

1. specified that all FFI and AD patients included in the study were homozygous for methionine at codon 129 of the *PRNP* (lines 160-166);

2. clarified the source of PMCA products used for inoculation (lines 160-168);

3. specified that PMCA products were collected at the third round of PMCA (lines 163-168, Figure legend 1);

4. included in each figure legend details about stains and antibodies used.

2. The discussion may also include the biosafety precautions for PMCA products, as well as recent studies of RT-QuIC products which appears to be a safer alternative for diagnostic purpose in light of a recent negative finding in OM of CJD.

We thank the reviewer for pointing this out and we have included in the discussion a paragraph that highlights the importance of manipulating infectious PMCA products by following the specific biosafety precautions used for brain material. The possibility to use the RT-QuIC as an alternative and safer diagnostic assay has also been mentioned (lines 696-699).

Reviewer #2:[…] The conclusions of the paper are supported by the data, but there are additional controls and information that could be included that would strengthen the manuscript.1. Based on the different sensitivities to proteolysis and guanidine denaturation among the brains of those mice inoculated with PMCA generated PrPSc versus those inoculated with FFI brain homogenates, the authors conclude that a bank vole adapted strain of FFI may have been generated. This conclusion could be strengthened by performing these same assays on the PMCA generated PrPSc prior to inoculation. This would demonstrate whether these biochemical differences existed prior to passaging in the BvPrP mice. In addition, it would be beneficial to assess the properties of the original FFI brain homogenate and how it may have changed when performing PMCA or passaging experiments.

We thank the reviewer for this useful suggestion. As indicated, we have performed additional proteolysis and guanidine denaturation analyses of the FFI brain homogenate (FFI-BH) and its product of amplification (FFI-BH_PMCA) prior to inoculation in BvPrP-Tg407 mice. Each sample has been analyzed in triplicate and the most representative Western blot images obtained from the analyses are reported in Author response image 1. Part of the results (the graphic representations and their detailed descriptions), have been included in the manuscript as Figure 4—figure supplement 2.

After three rounds of PMCA the FFI-PrP^Sc^ acquired distinct biochemical properties (especially with regards to Gdn-HCl stability) compared to those of the PrP^Sc^ present in the original inoculum. This might have also altered the infectious properties of the newly generated PrP^Sc^ that elicited neuropathological changes in BvPrP-Tg407 injected mice significantly different from those observed in FFI-BH challenged animals. Comparing the biochemical properties of PrP^Sc^ present in FFI-BH with those of the PrP^Sc^ generated in FFI-BH injected mice (especially in terms of PK resistance), we observed statistically significant differences which suggest that the PrP^Sc^ properties were not retained upon inoculation in mice. Similarly, the PrP^Sc^ present in FFI-BH_PMCA samples changed its biochemical properties upon inoculation in mice, in terms of both PK resistance and conformational stability. Thus, while some features of FFI-PrP^Sc^ changed either when amplified in vitro by PMCA or when challenged in vivo, other properties typically associated with this prion strain (e.g. the prevalence of the di-glycosylated PrP bands, its tropism for the thalamus and the presence of the un-glycosylated band migrating at 19kDa) were retained. As also mentioned in the manuscript, the same PK and Gdn-HCl analyses done for BH-FFI and BH-FFI_PMCA could not be performed for FFI-OM and FFI-OM_PMCA. This was due to the fact that the raw OM was already subjected to several analyses whose results have been published in a previous manuscript [see ref. 26 in the manuscript] and this sample was no longer available. An additional OM collection was not possible because the patient died after the first OM sampling. The amplified products were thoroughly used to perform BvPrP-Tg407 injection and the small amount left was not sufficient to perform neither PK (in triplicate) nor Gdn-HCl (always in triplicate) analysis. Unfortunately, the lack of raw OM did not allow us to perform additional PMCA analyses to generate new material useful for the analysis.Nevertheless, we believe that the PrP^Sc^ present in the olfactory mucosa might have been undergone the same selection/adaptation events observed for the PrP^Sc^ present in the brain when challenged in vitro or in vitro.

The new findings obtained after the analyses suggested by the reviewer were included and discussed in several sections of the manuscript, including results (lines 407-426), discussion (lines 617-625 and lines 659-665) and material and methods (lines 221-223 and lines 231-233).

2. The PrPSc deposition and glial pathology in Figures 6 and 7, respectively, are very difficult to see and compare among cohorts. Images with higher magnification or insets would be helpful.

According to the reviewer’s suggestion, we have prepared new Figures (6 and 7) by acquiring images at higher magnifications which better show the prion distribution and the glial activation in the brain of all FFI and AD injected BvPrP-Tg407 mice. This will facilitate comparison between cohorts. In the case of Figure 6, the figures related to the animals inoculated with FFI-OM_PMCA contain three insets useful to appreciate the presence and morphology of the plaque-like PrP^Sc^ deposits.

3. The authors include an observation in the discussion highlighting that one of the FFI-OM_PMCA animals had 2 distinct un-glycosylate PrP bands. This was not mentioned in anywhere in the Results section. Please discuss this observation in the appropriate Results section.

As suggested by the reviewer, we have described this finding also in the appropriate Results section (lines 347-351 and lines 362-364). The legend of Figure 3 has been modified accordingly.

Reviewer #3:The experiments are well-designed, logically organized, and convincingly show that PMCA successfully amplified FFI PrPSc. The amplified PrPSc was infectious to mice expressing BV PrPC, retaining some features of FFI PrPSc, but clearly selecting for a PrPSc conformer with differing biochemical properties than the original FFI prion, and thereby inducing a somewhat different pathologic profile. The experimental models are appropriate.The main issue lies in the novelty of the work and the scientific advance, since FFI infectivity in mice expressing bank vole PrPC has been shown, and FFI PrPSc has been previously amplified by PMCA.

We would like to thank the reviewer for his/her comments on our manuscript and take the opportunity to remark some of the novelties of this work. Although the FFI infectivity in mice expressing the bank vole PrP^C^ has already been shown, in our manuscript we have used a new transgenic model that generated two different PrP^res^ once infected with brain-derived FFI-PrP^Sc^ (one prion isolate that was more represented than the other). FFI prion might not be a pure strain and the possibility to study FFI with new in vitro or in vivo models will allow to investigate whether this strain is composed of a mixture of prion isolates, that can even undergo selection and adaptation phenomena according to the environment of replication (PMCA *vs* bioassay). This will have a significant impact especially in terms of disease treatment. Indeed, the use of therapeutic compounds, especially if administered in the preclinical stage of the disease and for long periods of time, might favor the generation of drug-resistant prion strains that can be responsible for the failure of the therapy.

We acknowledge that FFI-PrP^Sc^ has been previously amplified by PMCA. However, in our work, we were aimed at investigating whether and to what extend the OM amplified products could have retained (or modified) disease-specific biochemical or neuropathological features typically associated with the brain FFI-PrP^Sc^ potentially exploitable to perform an in vitro screening of compounds to identify their efficacy for each patient. Finally, we have highlighted that the PMCA generates infectious products and their manipulation/analysis require specific biosafety precautions.